# Kernel-DMD for multiome data integration and control

**Iro Pierides[1]\*, Hannes M. Kramml[1], Steffen Waldherr[1]\*, Wolfram Weckwerth[1,2]\***

**1** Molecular Systems Biology Lab (MOSYS), Department of Functional and Evolutionary Ecology, University of Vienna, Vienna, Austria, **2** Vienna Metabolomics Center (VIME), University of Vienna, Vienna, Austria

\* iro.pierides@univie.ac.at (IP); steffen.waldherr@univie.ac.at (SW); wolfram.weckwerth@univie.ac.at (WW)

## Abstract

Research in multiome data integration comes with the challenge of high-dimensionality and a small sample size in time series data. Traditional statistical tools often fail to capture true functional modules in large molecular networks, resulting in spurious associations. Dynamical systems theory overcomes this hurdle by assuming the biological system follows a trajectory that can be modelled in such a way that the interactions in the network have a causal nature and pertain to mechanistic processes. Here we use kernel-DMD, a data-driven dynamical systems tool for time series data, for multiome network integration in the exotic plant species *Clusia*. We uncover differing modes of photosynthesis that correspond to the C3-like or strong CAM dynamics of two species, *Clusia major* and *Clusia rosea* and implement a control strategy that enables the *in silico* phenocopying between the two species. We demonstrate the applicability of the Koopman operator to multiome data integration, uncover drivers of plasticity in molecular networks and also identify key biomarkers that could potentially establish more resilient forms of photosynthesis, such as CAM, for the introduction of new crop bioengineering possibilities in C3 plants.

## Author summary

High-dimensional multiome data are increasingly used to decipher molecular mechanisms in different complex biological systems. The challenge remains on how to extract useful information that is not only significant in terms of distributions but also in terms of the underlying causal dynamics. We evaluate the performance of kernel Dynamic Mode Decomposition (kernel DMD) in its ability to integrate multiome data and predict outputs of two *Clusia* species with different types of photosynthesis (C3-like and strong CAM). In this way we are able to decompose small time series data into dynamically distinct modules (eigenmodes) and provide a rank on the most significant features that are able to shift output

**Data availability statement:** The dataset is publicly available at Zenodo (DOI: 10.5281/zenodo.17370335). The data and code is also available on Github: https://github.com/IroPierides/kernelDMD-for-multiome-integration-and-control-.

**Funding:** This work was supported by FWF Austrian Science Fund. IP received funding from Austrian Science Fund with grant number: I 5234-B. HK received funding from Austrian Science Fund with grant number: DOC 111-B. The funders had no role in study design, data collection and analysis, decision to publish, or preparation of the manuscript.

**Competing interests:** The authors have declared that no competing interests exist.

distributions between the two species. Our results have impact in the fields of evolutionary biology and applied crop science in potentially switching to more efficient modes of photosynthesis. They also demonstrate the generalization and suitability of kernel DMD with control for complex multiomics datasets.

## Introduction

### Dynamic systems theory in biology

Network science has experienced an enhanced interest over the last years with a lot of research involved in understanding the complexity and causality within network interactions that drive many natural and artificial systems [1,2]. In biology, the measurement and collection of large-scale molecular data such as genomics, transcriptomics, proteomics and metabolomics [3,4] has imposed a growing need for new methods and tools to integrate and analyse multiomics interactions in ways that follow the real underlying mechanisms within molecular networks [5–7]. Often, purely statistically based machine learning tools are limited in their capacity to identify true mechanistic associations in the data due to relying on variances and correlations that might be spurious and not causal in nature. In contrast, tools that base their assumptions on a predefined model allow for a hypothesis on causality relations to be made that can be refuted or approximated based on prediction or reconstruction metrics. Dynamical systems theory is a mathematical domain that offers a plethora of such tools that contain a lot of generic formalisms, transferable within biological domains [8].

Widely used and well established in the fields of control engineering and physics, dynamical systems have also experienced a surge within biology from quantifying the dynamics of gene expression [9], to finding differential biochemical reaction kinetics [10], to generating distinct dynamical behaviours via toggle switches [11] and switching cancer cell types [12]. The plethora of applications acknowledge the dynamic nature of biological systems such as ecological networks, embryonic pattern formation and interactions within molecular networks [8]. Thus, there is a shift from static representations to increasingly dynamic accounts of biology [13,14]. This shift needs to be matched with a transition from the use of static statistical tools to dynamical systems tools that consider highly non-linear interactions in biology. Current challenges with ODE models for studying dynamical systems in biology is that the tools are mostly tailored for low dimensional systems [15] but usually ecological and multiome molecular data are high-dimensional (small sample sizes, large feature spaces) [8]. They also require a priori knowledge of kinetic equations with many unknown parameters [16] which are hard to define in high-dimensional biological systems.

### Dynamic mode decomposition as a data-driven dynamical system tool

Here we aim to explore the applicability of a data-driven dynamical systems tool, Dynamic Mode Decomposition (DMD) (see Methods), in the network integration of multiome short time-series data of two *Clusia* species, *C. rosea* and *C. major* that

exhibit alternate forms of plant photosynthesis. DMD is a well-known time series data-reduction approach, that approximates forward dynamics discretely in time [17] and is applicable to big datasets such as omics data with many unknown parameters. It was originally developed for decomposing high-dimensional data into coherent temporal components, also known as eigenmodes, that represent distinct linear trajectories in time. At the same time, it approximates the Koopman operator that allows to perform diagnostics of the lower-dimensional linear approximation model of the data. DMD combines the spatial data reduction step of principal component analysis (PCA) and time decomposition step of the discrete-time Fourier transform (DFT) [18]. Entirely based on measurement data and with no need to define an explicit physical model, it has been applied in many different fields such as fluid mechanics [19], climate and geosciences [20], biochemical systems [21], robotics [22], power grids [23] and economics [24]. The eigenmodes that are a result of DMD, can be causally linked to different behaviours of a system and provide physical interpretability, regime detection and classification. For example, DMD applied to functional magnetic resonance imaging (fMRI) scans resulted in clusters of eigenmodes that corresponded to resting state networks (RSNs) of different brain regions [25]. Furthermore, the linear Koopman operator can be used as input to optimal control strategies that steer the direction of the system to a desired outcome [26–28].

Data-driven approaches have emerged as effective tools to bypass the need of assumption-based forward models and corresponding parameters [8,29,30]. We previously implemented a data-driven inverse modelling approach using the stochastic Lyapunov matrix equation to reveal control points in highly complex metabolomics data by Jacobian reconstruction [8,29,31]. This allowed the identification of a novel metabolic checkpoint for macrophage polarization [32] which eventually was proven to control tumor-associated macrophages and tumor growth [33,34]. Generic principles of the inverse stochastic Lyapunov equation can be applied universally in biology and ecology [8]. These methods are fundamental for understanding biological systems because they reconstruct realistic models directly from empirical data, minimizing the reliance on an a priori parameterized model [8]. By using DMD, we overcome certain limitations of the Jacobian reconstruction while extending to non-linear and higher order dynamics (see kernel-DMD in Methods) within the bounds of the sampled time series of a dataset. With the combinatorial approach we use here, more than one omics data type can be integrated into dynamic network representations without the need of *a-priori* knowledge of network interactions as required for the Jacobian reconstruction. The method also extends to bigger datasets as found within transcriptome or proteome data due to its data reduction step, for which the Jacobian derivation becomes more unstable with increasing feature size [35]. While the Jacobian operates on steady-state data using covariance matrices across different conditions, DMD is applied on time series and discovers time-dependent dynamics. Therefore, DMD for multiome network integration is a significant extension of the data-driven dynamic toolset.

A limitation of using dynamical systems tools, or any tool borrowed from different scientific disciplines, is whether they are appropriate in providing a holistic mechanistic understanding of molecular systems for which the governing equations are not explicitly known [36]. However, even though a chosen model might not represent the full reality of a biological system, it can at least provide knowledge of some of its properties such as stability, modularity, bifurcation parameters or emergence and is therefore suitable for specific applications such as input control. Within the kernelized form of DMD, the type of nonlinear interactions between variables can be approximated using a kernel. Here we used a kernel-DMD variant as introduced in LANDO [37] which overcomes the limitations of linear DMD for nonlinear systems such as multiome networks. The kernel enables implicit physical assumptions to be made in the interactions of features via the use of the dot product which can be embellished within simple tailored functions such as the polynomial or gaussian functions. Various kernel-DMD approaches have been developed [38–41] and evaluated in their application to various nonlinear systems that showed better performance and Koopman spectra than classical DMD [40,42–45]. We chose LANDO because it circumvents the need to explicitly define observables in the higher-dimensional space, such as SINDy [38], which is computationally expensive. Instead it scales the problem to the dimension of the time component of the series and not feature size, which is particularly fitting to molecular data with high feature size and low sample size. Contrary

to other kernel-DMD efforts, LANDO maps the kernel and its learned weights to the original state values, which enables the extraction of the linear Koopman operator in the original feature space providing more physical interpretability of the feature interactions in the Koopman network. This is helpful for concrete conclusions to be made in terms of molecular networks. It also disambiguates the linear from non-linear components, which we showcase in this paper to be useful for different downstream applications such as the eigendecomposition emerging from the linear part and control mechanisms from the nonlinear part. Here we take advantage of the kernel method, to tailor a "Hill function"-like kernel, via a combination of sigmoid and linear kernels (see Methods section below). Hill function dynamics are frequently used to model molecular network interactions [46] and the hypothesis is that they are also appropriate for modelling multiome data interactions. The use of a coarse-grained kernel DMD data-driven model, is an efficient way to circumvent the problem of high-dimensionality in datasets where an interaction model between all sets of features is impossible to know a priori.

## Application to differential fluxes between different types of photosynthesis

We demonstrate the suitability of multiome Koopman integration by identifying dynamical differences between two types of plant photosynthesis (Crassulacean acid metabolism (CAM) and C3-like, S1 Fig). CAM is a type of photosynthesis that has physiologically evolved in adaptation to low water availability. During CAM, stomatal opening occurs in the night instead of the day, resulting in nocturnal uptake of $CO_2$. Phosphoenolpyruvate carboxylases (PEPCs) carboxylate Phosphoenolpyruvate with $CO_2$ eventually stored as organic acids such as malate or citrate within the vacuole. During the day decarboxylation of stored organic acids restore $CO_2$ levels, thereby ameliorating the need for stomatal opening and minimizing water loss. *C. rosea,* a well-known constitutive CAM species [47], differs from its close relative, *C. major,* which is a C3-like plant.

Here, we aim to uncover the differences in the subtypes of photosynthesis within these two plant species using our combinatorial Koopman approach for network integration (see workflow Fig 1). Firstly, we evaluate different kernels (Fig 1A2) and their hyperparameters to evaluate whether the sigmoidal or "Hill function"-like kernel is a good fit to both forward dynamics and output measurements in a multi-objective function. We use the Taylor series expansion to extract the linear and nonlinear parts of the model. The linear part is effectively the Koopman operator (Fig 1A3) of which the eigendecomposition results in eigenmodes that correspond to different temporal dynamics. The distinction between weak versus strong CAM photosynthesis becomes clearer once we relate the predictive capacity of selected eigenmodes to stomatal opening $CO_2$ gas exchange measurements – a distinctive physiological signature between C3 vs CAM due to opposing stomatal opening trends (Fig 1A4). We also introduce new input control mechanisms in *C. major* for switching between photosynthetic types and phenocopying the *C. rosea* output signature (Fig 1B). This comprises of several steps starting from the identification of appropriate control matrix and inputs making use of the nonlinear part of the learned model (Fig 1B1), to reducing the dimensionality of the data (Fig 1B2), carrying out alignment (Fig 1B3) and then performing linear Model Predictive Control (MPC) in the reduced subspace (Fig 1B4). The linearization of the system first by Taylor series expansion to get the Koopman operator and then SVD of the nonlinear part of the Taylor series to obtain the linear control matrix *B*, was crucial for making the system amenable to linear MPC, which requires a linear model [48]. The data reduction step was also an important prerequisite to linear MPC because the latter failed to predict good output reconstruction accuracies in the full feature space. It was therefore imperative to find a method that selected the most important features in terms of dynamics in order for MPC to work efficiently with regard to time and computational complexity [49]. We also found that the alignment step through the use of a transformation matrix was also a necessary prerequisite to MPC for proper alignment of Koopman matrices between the two systems that at the same time could find the right projection between output spaces.

To our knowledge, our work is one of the first attempts to validate kernel-DMD with subsequent optimal control in a high-dimensional multiome dataset and its capability for phenocopying. Our methodology has impact in the fields of crop science that aims to increase water use efficiency of C3 crops under arid, harsh conditions [50] by introducing new

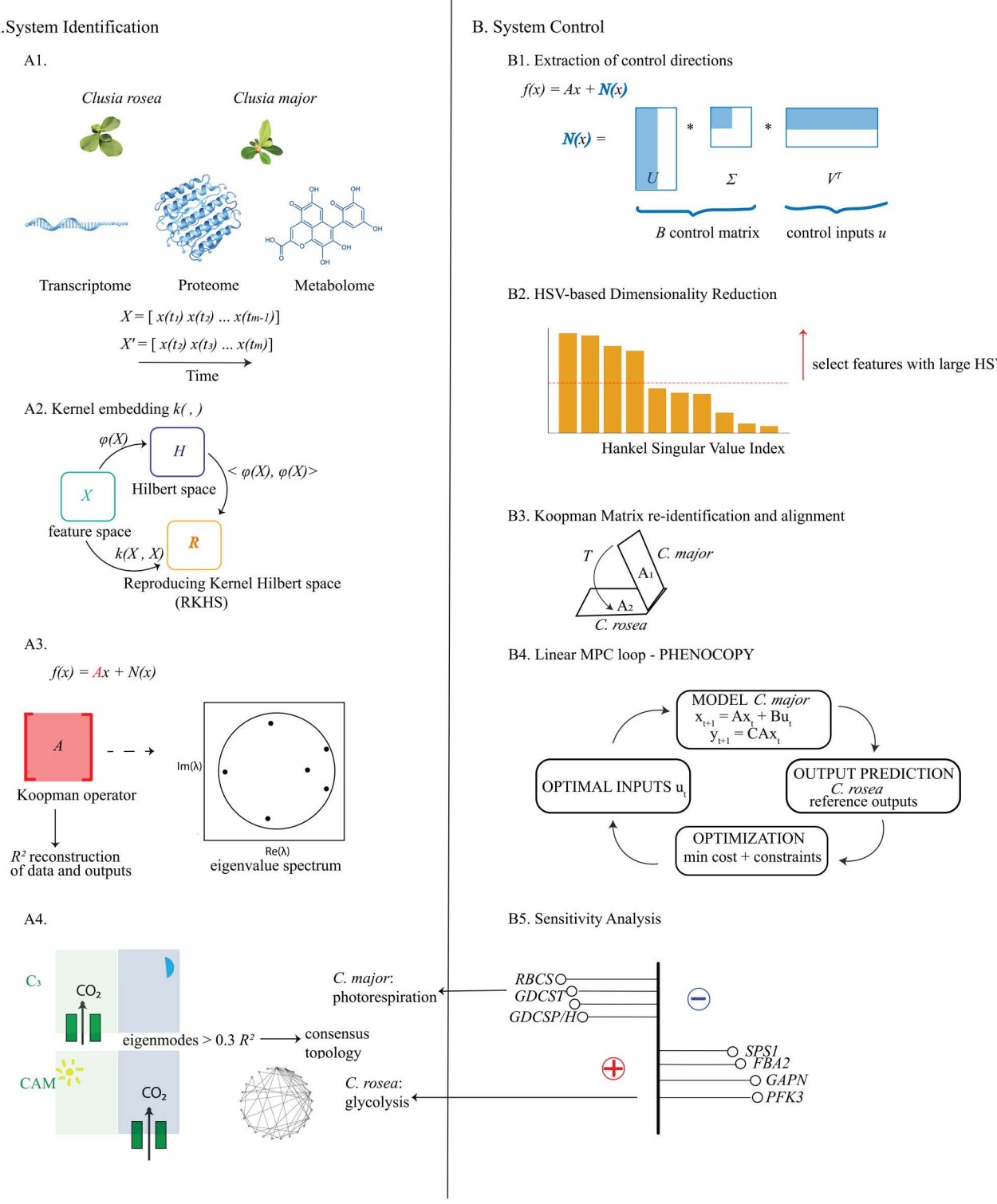

**Fig 1. Illustration of kernel-DMD with control workflow applied to multiome timeseries data of two *Clusia* species. A)** System Identification using kernel-DMD. A1) Multiome timeseries dataset collection (transcriptome, proteome and metabolome) of two *Clusia* species *C. major* and *C. rosea* with different photosynthetic physiotypes (C3-like vs strong CAM respectively). Time snapshot rearrangement for input to kernel-DMD where *X* includes time intervals 1 to *m*-1 and *X'* one time step ahead (time interval 2 to *m*) with *m* the total number of time points. A2) The kernel applied to the concatenated

multiome $X$, implicitly embeds the original features in the Reproducing Kernel Hilbert Space (RKHS) which includes non-linear interactions. This way we avoid an exhaustive search for optimal parameters of inner products in the higher-dimensional nonlinear feature space $\phi(x)$. A3) Linear Koopman model extraction results in coherent dynamics described by the eigenvalues of the Koopman operator. The model is evaluated based on the linear data and output reconstruction accuracies ($R^2$ score). This enables a hyper-parameter search for optimal kernel choice and its parameter values in step A2. A4) After fitting kernel--DMD to both species, their eigendecompositions are compared and the eigenmodes that best predict $CO_2$ gas exchange data (as a physiological signature of opposing CAM trends) are stored. Their most frequent edges are retained into a consensus eigenmode topology for which a glycolysis module is more dominant in *C. rosea* and a photorespiration module in *C. major*. **B)** System Control for the phenocopy of CAM output signature as exhibited in *C. rosea* within a C3 molecular background as in *C. major*. B1). The Taylor series expansion of the kernel-DMD model disambiguates the linear from nonlinear parts. The nonlinear part $N(x)$ can act as a forcing constraint to the linear part. Its reduced control basis via Singular Value Decomposition (SVD) leads to the $B$ control matrix and control inputs $u$. B2) Hankel Singular Values are used for feature selection as a way to select the most observable and controllable features for data reduction. B3) System re-identification in the reduced feature space is followed by projection of the *C. major* Koopman operator to that of *C. rosea* while at the same time ensuring alignment of their output dynamics. B4) The aligned linear Koopman matrices are used as input to Model Predictive Control where phenocopy is achieved for predicting new outputs in *C. major* that closely resemble those in *C. rosea* by finding optimal control inputs $u$. B5) Sensitivity analysis identifies positive feedback applied to glycolytic enzymes and negative feedback applied to photorespiratory enzymes. All images in Fig 1 were drawn by hand, except the photos of *C. major* and *C. rosea* in Fig 1A which were taken by colleagues who gave full permission to publish them under the CC BY 4.0 license. Images of the transcriptome, proteome and metabolome in Fig 1A were produced by ChatGPT5.

emerging bioengineering possibilities of advanced mechanisms of photosynthesis such as CAM. With this approach we also ameliorate limitations of limited sampling points obtained during typical measurements in ecological and biological systems. DMD discretely maps each time snapshot to the next without the requirement of significant time lag within the time series. Below we outline the methodology and workflow that uses the DMD formulation as a backbone.

## Methods

We now provide the mathematical workflow that is a solution to the goal of identifying the system dynamics that differentiate the two *Clusia* species via their eigenmode decompositions as well as control mechanisms that introduce a phenocopy of CAM outputs of *C. rosea* within *C. major*. The pipeline (Fig 1) follows the main following steps: 1) System identification of each species via kernel-DMD (Fig 1A2), 2) comparison of the consensus eigenmodes of the linear Koopman operator that best predict the $CO_2$ gas exchange data between the two species (Fig 1A4), 3) System control by identification of the control matrix $B$ (Fig 1B1), 4) data-reduction via Hankel Singular Values (HSVs) (Fig 1B2) and system reidentification, 5) system alignment and transformation between the two reduced systems (Fig 1B3), 6) the phenocopy of *C. rosea* outputs in *C. major* via linear Model Predictive Control (Fig 1B4) and finally 7) identification of multiome biomarkers for control with positive and negative feedback based on the sensitivity gain matrix (Fig 1B5).

### Data and data preprocessing

Our dataset is high-dimensional (23 timepoints and 390 features) and includes 3 omics types: transcriptome, proteome and metabolome. Throughout the paper we make comparisons between C. *major* (represents C3-like photosynthesis) and *C. rosea* (represents strong CAM photosynthesis) (Fig 1A1). Transcripts with a TPM value below 1 were filtered out as well as low variant features (variance threshold $<0.3$) in the proteome and transcriptome. All metabolites and CAM-related features according to our previous pathway annotation, were included in both proteome and transcriptome as well as the features corresponding to highest loading scores of the first six Principal Components after PCA data reduction. Any features not found in a species were 0-padded. Protein and transcript data retained were in equal proportions while the metabolome was less represented in terms of metabolites measured. After addition of a small random noise in each feature column to extend the timeseries from 12 to 24 samples, the concatenated multiome was z-transformed around mean 0 and standard deviation 1. The samples were rearranged such that the timeseries was in consecutive timesteps (4am, 8am, 1pm, 7pm) with replicates serving as distinct points in the series.

## Part I: System identification

### Dynamic mode decomposition and the Koopman operator

We consider dynamical systems that can be expressed as ordinary differential equations as in

$$\frac{d}{dx}x(t) = F\left(x(t)\right) \tag{1}$$

where $F$ describes the continuous-time evolutionary dynamics of an $n$-dimensional vector $x$ that characterizes the state of the system at time $t$, more specifically the concentrations of concatenated multiome features after z-transformation. $F$ gives information on further tasks such as physical understanding, prediction and control. Due to the high non-linearity of dynamic systems, one major goal is to identify a new vector of coordinates $z = \phi(x)$, such that the time-evolution can be represented in linear terms. In practice, data is sampled discretely in time, so the aim is to find a matrix $A$, known as the Koopman operator [51,52], that advances dynamics discretely in time as in

$$x_{t+1} = Ax_t. \tag{2}$$

The eigendecomposition of the Koopman operator $A\phi\left(x_k\right) = \lambda\phi\left(x_k\right)$ completely characterizes the dynamics of the linearised system.

In its simplest form, Dynamic Mode Decomposition (DMD) is a data-driven method that approximates the Koopman operator from discretely sampled data. The snapshot data is arranged into two matrices $X$ and $X'$:

$$X = \left[x\left(t_1\right) \; x\left(t_2\right) \ldots \; x\left(t_{m-1}\right)\right]; \; X' = \left[\; x'\left(t_2\right) \;\; x'\left(t_3\right) \ldots \;\; x'\left(t_m\right)\right] \tag{3}$$

where $m$ is the number of time steps. The Koopman DMD approximation in terms of these matrices is

$$X' \approx AX. \tag{4}$$

The best fit $A$ that advances snapshots linearly forward in time can be formulated as an optimization problem

$$A = \arg\min_{K} \|X' - AX\|_F = X'X^{\dagger} \tag{5}$$

where $\dagger$ denotes the pseudo-inverse. The pseudo-inverse can be computed using Singular Value Decomposition (SVD) as $X = U\Sigma V^T$. In practice, the effective rank of the data matrices after SVD is lower than their full dimension, so $A$ can be projected to a lower dimension using the first $r$ Principal Orthogonal Decomposition (POD) modes of $U$, approximating the pseudo-inverse using the rank-$r$ SVD approximation $X \approx U_r\Sigma_r V_r^T$:

$$\widehat{A} = U_r^T A U_r = \; U_r^T X' X^{\dagger} U_r \; = U_r^T X' V_r \Sigma_r^{-1} U_r^T U_r \; = U_r^T X' V_r \Sigma_r^{-1}. \tag{6}$$

Spectral properties of the Koopman operator give a complete diagnosis of the dynamical system with the description of the dynamic flow by the linear combination of dominant coherent structures, i.e., its eigenmodes [17]. The Koopman operator gives an understanding of the general parameters and model equations that govern the system using only data as input. This is especially useful in biology, where models are not readily available or are simply assumed and could further facilitate understanding of evolutionary adaptive mechanisms as well as their control [53].

## Kernel DMD for multiome network integration

While standard DMD is a linear method to identify the underlying forward dynamics $F$ in (1), it cannot fit well to highly non-linear systems as found within molecular interaction networks. Relatively improved DMD algorithms such as extended DMD (eDMD; [39]) or sparse identification of nonlinear dynamics (SINDy; [38]) can disambiguate between linear and non-linear dynamics but do not scale well to highly dimensional systems such as omics datasets. More recently, [37] developed LANDO (Linear and Nonlinear Disambiguation Optimization) that both applies well to non-linear and high-dimensional systems via the use of kernels. Kernels are continuous functions $k : \mathbb{R}^n \times \mathbb{R}^n \to \mathbb{R}$ that combine features as inner products $k(X_i, X)$ within a given Hilbert space $H_k$ where there is a mapping $\phi : \mathbb{R}^n \to H_k$. The true dynamics $F$ (1) can be modelled using a model $f(x)$ by letting

$$f(X) \approx \sum_{j=1}^{N} \xi_j \phi_j(X)$$

(7)

where $\phi$ is a candidate model term for every feature in $X$ and $\xi$ is the coefficient that determines how active is each $\phi$. For large feature dimensions $N$, finding suitable terms in (7) becomes computationally expensive and the optimization problem is very hard to solve. The problem (7) can be rewritten in terms of a linear combination of kernels

$$f_i(X) \approx g(X) = \sum_{j=1}^{m} w_{ij} k(X_j, X)$$

(8)

where $m$ is the total number of training samples and $w_j \in \mathbb{R}^n$ the weight vectors [37]. The problem now scales over the number of snapshots $m$ instead of the number of features, which makes it more tractable to solve. The full equation of dynamics is

$$X' = g(X) + \vartheta R(w)$$

(9)

where $g(X)$ is an eigenfunction (8). We set $\vartheta R(w)$ as the Elastic Net regularization for constraining the weights $w$ as in:

$$\vartheta R(w) = \alpha(1 - \rho) \; \|w\|_1 + \rho \; \|w\|_{fro}$$

(10)

with $\alpha = 1$ (penalty coefficient) and $\rho = 0.99$ (ElasticNet mixing parameter that ensures uniqueness of solution). We also use this type of regularization in subsequent steps. Elastic net regularization has been proven helpful as a constraint in other DMD studies [54] to promote sparseness and was more effective for our dataset compared to just $L_1$ norm regularization. The value $\alpha$ scales the strength of regularization strengths, with larger values prioritizing sparsity terms preventing overfitting. Values of $\alpha$ below 0.01 lowered the output reconstruction accuracies during the phenocopy of CAM, but did not affect data reconstruction accuracies.

The problem in (8) reduces the parameter search for $g$ to a linear search for weights $w$ while combining the original features in $X$ using non-linear kernel functions in $k$. This method is also suitable for high-dimensional systems, like our dataset, since it scales well with the number of samples and not the feature size. We evaluated the performance of 4 different kernels (linear, gaussian, polynomial and sigmoidal, Table 1) using data reconstruction accuracy (coefficient of determination, $R^2$) to see which one best approximates the dynamics of multiome networks. For each one, we carried out a hyperparameter search by varying their respective parameters (Table 1). The linear kernel is the dot product between features and computes all linear combinations of the feature input space. The gaussian kernel can be interpreted as defining the characteristic length scale of features in $x$. Thus, concentrations between features that have similar distributions have higher values in the gaussian kernel, while trajectories that are too disparate between them will have almost 0 value. The polynomial kernel computes the degree-$d$ polynomial between features and thus considers quadratic, cubic and higher

**Table 1. Kernel functions, their respective gradients and hyperparameters.**

| Kernel type | Function $\mathbf{k}\,(X_j, X)$ | Gradient $\nabla \mathbf{k}\,(X_j, X)$ | Hyperparameter | Literature |
|---|---|---|---|---|
| linear | $x^T x'$ | $x$ | | [37, 55-57] |
| Gaussian | $\exp\left(-\frac{\|x - x'\|^2}{2\sigma^2}\right)$ | $-\frac{1}{\sigma^2} diag\left(\|x_j - x\|_2 \exp\left(-\|x_j - x\|_2^2 / (2\sigma^2)\right)\right)$ | $\sigma$ | |
| polynomial | $(\gamma\, x^T x' + c)^d$ | $d(\gamma\, x^T x' + c)^{d-1} x^T$ | $d, \gamma, c$ | |
| 'Hill function' (sigmoidal) | $c_s * \tanh\left(\gamma\, x^T(I - L)\, x' + c\right) - c_l * x^T x'$ | $c_s * \gamma \left(1 - \tanh^2\left(\gamma\, x^T(I - L)x' + c\right)\right) x - c_l * x$ | $\gamma, c, c_s, c_l$ | |

$\sigma$ is a normalization coefficient, $d$ the degree of the polynomial, $\gamma$ the coefficient for the vector inner product (i.e., the strength of interacting features) and $c$ a constant coefficient, $x$ the concatenated input data and $x'$ the mean of $x$ that serves as a fixed point (base state). $I$ is the identity matrix and $L$ the graph Laplacian.

order interactions between features. This kernel allows for highly non-linear interactions between features that can also be found in Michaelis-Menten dynamics [58]. Finally, we also consider the sigmoidal kernel for inter-omics interactions, which in combination with the linear kernel as a degradation term can approximate Hill function dynamics of the form $h(x) = \frac{x^n}{k^n + x^n}$ which are often used to model molecular networks (see [59] for a full derivation and S1 Appendix).

Within the sigmoidal kernel we also implemented the graph Laplacian $L$ (see Table 1) for integration of a-priori knowledge of molecular pathways in terms of protein-protein (PPI) and protein-metabolite interactions. Similar to FMvPCI [60], which integrates prior knowledge of protein–protein interactions (PPI) via a multiview fusion approach, our method incorporates PPI data obtained from the STRING database and protein-metabolite interaction information from KEGG, allowing for better-informed dynamic representations of molecular networks. As demonstrated by [61] in their work on Link-Based Attributed Graph Clustering, combining structural information and node attributes significantly improves clustering accuracy in complex networks. This approach helps overcome challenges related to the noise and incompleteness of biological interaction data. An adjacency matrix $M$ was constructed for the retained input data after PCA data reduction and a threshold of 0.9 confidence score was used to retain the most significant interactions. The graph Laplacian $L$ was found by

$$L = D - M \tag{11}$$

where $D$ is the degree matrix obtained by setting the sum of incoming edges for each row entry of $A$ in each of diagonal. The Laplacian encodes the local connectivity of data points in the graph. By incorporating it within the dot product of the sigmoidal kernel, the global and local structure of the data is included in the similarity measure [62,63]. The kernel will now be sensitive not only to the raw similarity of feature vectors but also to the relationships encoded in the graph, which capture higher-order dependencies between data points.

## Multi-objective function for model weights

For the combination of all omics subsystems into a single linear operator, we concatenate them into one input array where each subscript corresponds to a different subsystem. We apply each kernel to the concatenated multiome and map it forward in time. At the same time, we map inputs to physiological outputs $Y$ in the following multi-objective function:

$$\min_{w_A, w_C} \; \|X' - w_A k(X, X)\|_{fro}^2 - \|Y - w_C k(X, X)\|_{fro}^2 + \vartheta R(w_A, w_C) \tag{12}$$

where the dash in $X'$ corresponds to one timestep forward as in (3), $X$ are the multiome inputs and $Y$ are relevant output data that differentiate between strong and weak CAM as defined by common knowledge (see below). $w_A$ are the weights that map the kernel of the multiome inputs $X$ forward in time to $X'$, while $w_C$ are the weights that make the

kernel mapping to the outputs *Y*. These weights are crucial because they determine how much influence each kernel function has on the optimization, essentially controlling the model's fit to the data. The weights are learned through approximating Bayesian posterior distributions using a linearized approach (solver 'CLARABEL', *cvxpy* in python) where the objective function is minimized iteratively. At each step, the optimization adjusts the weights in order to reduce the overall error between the predicted values (based on the weighted kernel) and the target values. The representation theorem in kernel learning [64], which states that a nonlinear function can be represented as linear functions in a higher dimensional feature space, ensures that the kernel can be mapped linearly forward in time via the use of these weights.

Solving the multi-objective optimization problem (12) means finding a subset of eigenfunctions of each subsystem $X_i$ (where *j* is the variable index corresponding to each of the three omics data types, e.g., 1 for transcriptome, 2 for proteome, 3 for metabolome) that are simultaneously required to capture the output dynamics *Y*. Because we combine both objectives in one equation, the Koopman dynamics, eigenvalues, and eigenmodes are constrained to distinguish explicitly between the two types of photosynthesis captured by differences in *Y*. To form the concatenated outputs in *Y* we chose malate, gas exchange measurements $H_2O/CO_2$ (mM m$^{-2}$ s$^{-1}$/ μM m$^{-2}$ s$^{-1}$), Phosphoenolpyruvate Carboxykinase 1 (PCK1; OG0002610::H1), Phosphoenolpyruvate carboxylase 1 (PEPC1; OG0001285::H1), alpha-glucan phosphorylase 2 (PHS2/ PHO1; OG0004358::H2) and pyruvate phosphate dikinase (PPDK; OG0001386::H1) as distinguishing CAM signatures between the two species. PHS2/PHO1 and PPDK are involved in plastidic starch recycling and carbon breakdown pathways, which as mentioned before, were found to contribute to differential evolutionary CAM trajectories in the *Clusia* species, based on a genome pseudogenization analysis. All outputs in *Y* are z-scored. By finding suitable weights $w_A$ and $w_C$ we construct our model *F(x)* (1) by mapping nonlinear interactions within the multiome data *X* discretely forward in time (i.e., to *X'*) and to output measurements (*Y*). The idea is to relate the dynamics learned from the Koopman operator to outputs *Y*, by evaluating how each eigenmode reconstructs each output (see below in Methods). In this way there is a spatiotemporal link of the original feature space to each output and its relation to CAM photosynthesis via the eigendecomposition of the multiome dynamics. We also use the same types of outputs *Y* for the phenocopy of the output signature of *C. rosea* within *C. major* (see Methods below) showcasing the ability of our control pipeline in shifting C3 output measurements to CAM trends.

**Linear Koopman operator from Taylor series expansion**

The learned model $w_A k(X_j, X)$ (12) captures the full forward dynamics but is implicit in nature: it doesn't provide adequate interpretation of the physical interactions in the system. For this we need to extract and disambiguate the linear from nonlinear dynamics of the model by using a Taylor series expansion [37] around a base state $\bar{x}$ which is taken as the mean of the input states over all features, as in

$$F(x) = F(\bar{x}) + \nabla F(\bar{x})^T (x - \bar{x}) + \frac{1}{2}(x - \bar{x})^T \nabla^2 F(\bar{x})(x - \bar{x}) + \ldots \tag{13}$$

The linear operator $A = \nabla F(x)_{x=\bar{x}}$ consists of the partial derivatives of the rate of change around base state $\bar{x}$ which is also the gradient, or the linear Koopman operator. Since the model is composed of linear combinations of kernels the gradient can simply be expressed as

$$\nabla F(x) = w \nabla k(X_j, X) \tag{14}$$

The respective kernel gradients can be found in Table 1. The Koopman operator *A*, can be interpreted as a multiome network, where the nodes are the features of the original feature space and edges are the magnitude of rate of change

from one node to the other. Thus, we begun with a full kernel model of the forward dynamics and ended with the identification of the linear Koopman operator, which is the final goal of any variant of DMD.

### Eigendecomposition of Koopman operator

The SVD decomposition of the kernel gradient $\nabla k\left(x_j, x\right) = U\Sigma V^T$ leads to the reduced linear Koopman operator $\hat{A} = U^T A U$ by projection of the full Koopman operator $A$ onto the columns of $U$. If the eigendecomposition of $\hat{A}$ is $\hat{A}\Psi = \lambda\Psi$ then

$$\Psi = \frac{1}{\lambda}WV\Sigma\Psi \tag{15}$$

are the eigenmodes of the system, where *W are the learned weights*. The eigenmodes (15) of the Koopman operator have a clear physical interpretation where the eigenvectors $\Psi$ are the spatial modes and represent the temporal components [37]. Stable constant temporal dynamics can be found within eigenmodes that have eigenvalues close to the unit cycle and can correspond in the case of biological systems to maintenance of homeostasis or dynamics that need to be sustained within the diurnal cycle of the plant. Decaying eigenmodes have eigenvalues that lie within the unit cycle (absolute values less than 1) and can be part of pathways that are active in one part of the day and then decay in another. Oscillating dynamics (like the circadian clock) can be aligned with eigenmodes that have imaginary parts in their eigenvalues, and these come in conjugated pairs. The combination of eigenmodes completely describe the dynamics of the linearized system represented by the Koopman operator and can be used to functionally distinguish two dynamic systems in terms of their temporal components. Each eigenmode has a rank of absolute values of each feature in the original feature space and this provides a natural clustering of the network nodes. Features with higher ranks in one mode can contribute more to its type of dynamics than within others. We chose to retain the number of eigenmodes equal to the rank of the kernel's gradient singular values with a cumulative sum ratio of more than 0.9, which represent the most energetic modes (Fig 2).

We associated the qualitative nature of each eigenmode using the modal reconstruction accuracies of CAM-related outputs (Fig 3). In order to make valid conclusions on the qualitative nature of eigenmodes, we chose to average results over many iterations selecting those eigenmodes that predicted $CO_2$ gas exchange outputs well ($R^2$ score $> 0.3$), since stomatal conductance was the main physiological difference between C3 and CAM dynamics in *C. major* and *C. rosea* respectively. In particular, for each iteration the weighted graph $G$ of the selected eigenmodes, for which only the first 60 features with highest ranking absolute values were kept, was calculated as an outer product [65]

$$G = \Psi\Psi^T \tag{16}$$

keeping edges with absolute weight above the upper quantile. The retained edges were added after each iteration to a consensus graph and edges with frequency above 0.5 were kept (S1 and S2 Tables).

## Part II: System control

### Extracting the control input matrix from Taylor series residuals

Since CAM is a photosynthetic mode with higher water use efficiency, one aim was the identification of optimal control that shifts dynamics from C3 levels (as in *C. major*) to strong CAM (as in *C. rosea*) dynamics (Fig 1B). To this end, the nonlinear higher order terms $N:\ \nabla^2 F(x)_{x=\bar{x}}$ from the Taylor series expansion (13) act as a nonlinear forcing to the linear operator $A$ and can be used to identify control matrices. $N$ is found by

$$N(x) = F(x) - F\left(\bar{x}\right) - \nabla F(\bar{x})^T\left(x - \bar{x}\right) \tag{17}$$

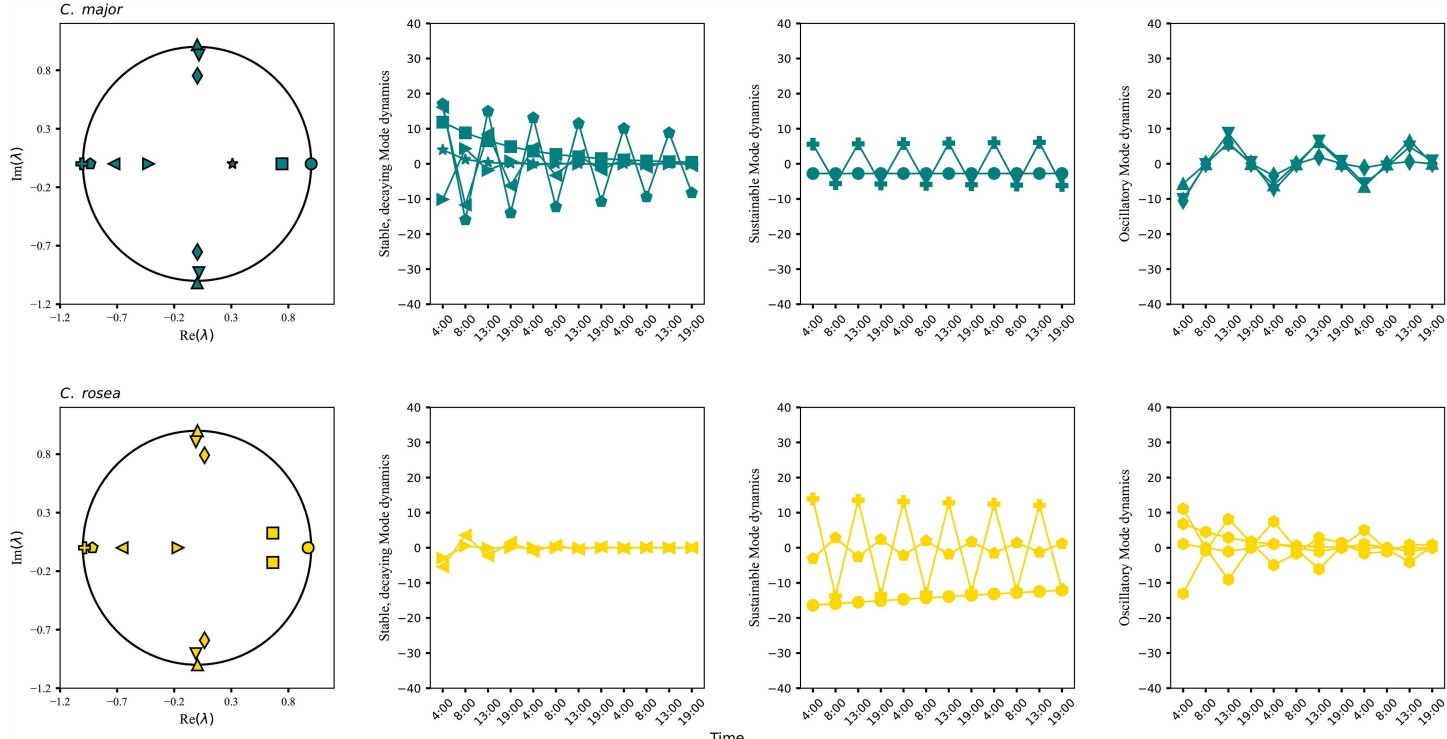

**Fig 2. Eigenmode decomposition of the two *Clusia* species.** The eigenvalues and their respective eigenmode amplitudes (same shape) are shown for *C. major* (top row) and *C. rosea* (bottom row) from one example iteration. The eigenmodes can completely characterize the decomposed dynamics of the system. Eigenvalues within the unit cycle describe stable eigenmodes with decaying dynamics, eigenvalues near the unit cycle are for sustainable dynamics and eigenvalues with imaginary parts for oscillatory dynamics. Each eigenmode has a feature rank which describes its qualitative nature. This is an example from one iteration of the kernel-DMD algorithm which introduces some variability in the eigenmodes due to stochastic initialization and additive noise added at each iteration. Overall eigenvalues and eigenmode amplitudes are expected to follow trends as shown in this figure.

which is the nonlinear residual term after subtracting the base state (input mean) and linear (gradient) dynamics from the whole state dynamics *F(x)* [37]. In order to extract meaningful information from *N*, it was linearly decomposed by SVD yielding left singular vectors that span the image (output or column space) of the nonlinear effects. These vectors represent the dominant directions or modes of nonlinear forcing in state space where it is projected or acts on. Multiplying the left singular vectors by the singular values after truncation gives

$$B = \hat{U}\hat{\Sigma} \qquad (18)$$

where *B* is the control matrix (Fig 1B). Scaling $\hat{U}$ by singular values $\hat{\Sigma}$ amplifies the effect of the dominant nonlinear modes according to their singular values. The cut-off value for truncating *Σ* was chosen as the number of singular values whose cumulative sum exceeded 90% of the total sum of all singular values, explaining more 'energy' in the nonlinear part. Conversely, the right singular vectors after SVD of *N(x)*, give time coefficients of the contribution of each nonlinear mode in *U* across time.

## Data reduction using Hankel singular values

Hankel Singular Values were used for feature selection in order to reduce the system to a smaller subspace in terms of dynamics. This required the calculation of the controllability and observability Gramians that make use of the linear

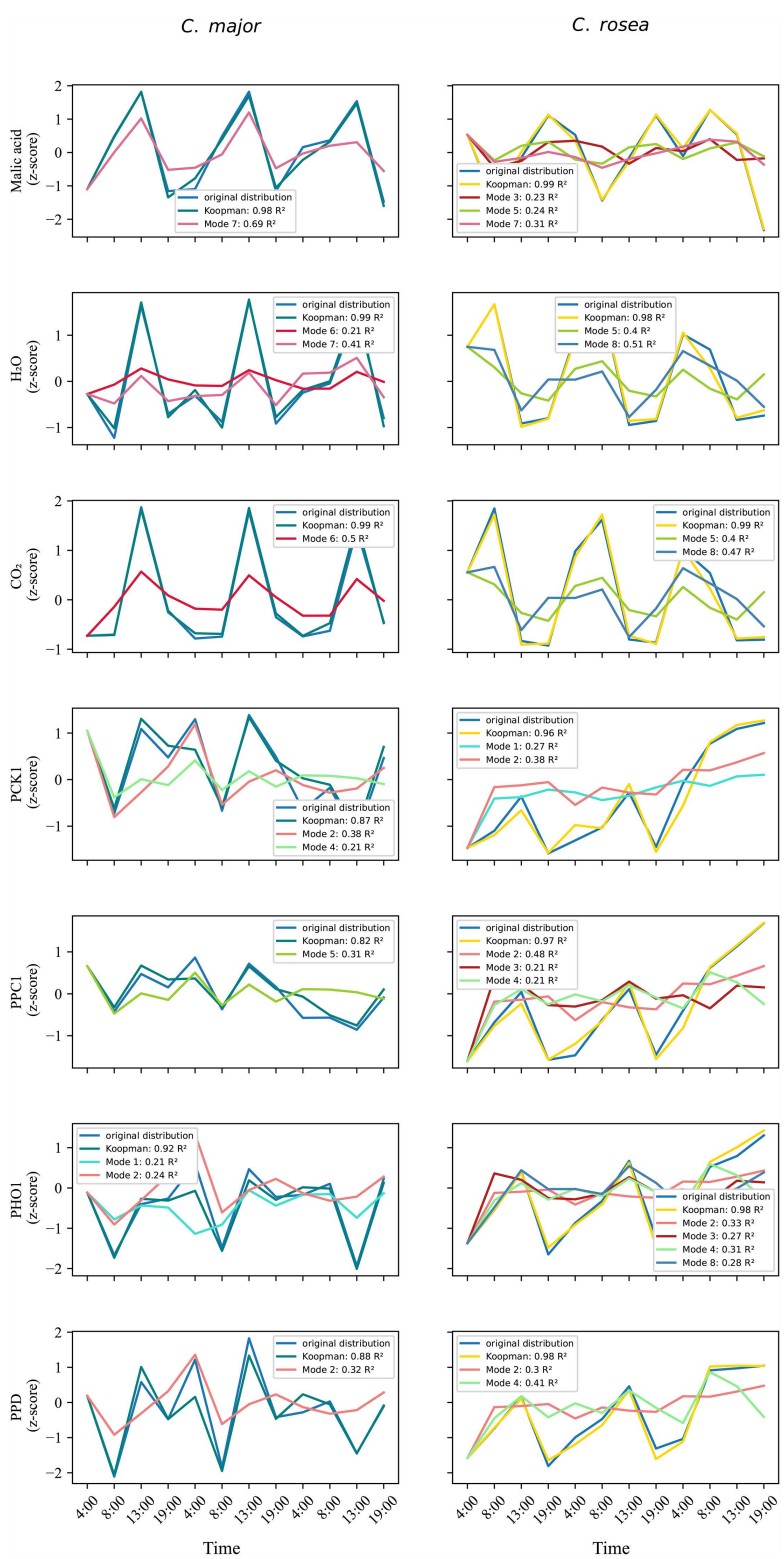

**Fig 3. Koopman and eigenmode phenotype reconstructions.** The fit of the kernel was evaluated based on Koopman and eigenmode reconstruction accuracies ($R^2$ score) of the phenotypes. The eigenmodes that predicted each output well distinguish the two species based on their feature ranks and temporal dynamics. In this way the whole multiome network can be decomposed into its temporal constituents which characterize different system

outputs that relate to the different forms of photosynthesis between the two species. Due to the variability in the eigendecomposition introduced by stochastic initiation and resampling in kernel-DMD, the figure shows an example of how each eigenmode relates to the outputs. This can slightly change between iterations and therefore we do not elaborate on the biological significance of modal reconstruction accuracies from this particular iteration.

operators $A$, $B$ and $C$ derived in previous steps. The controllability Gramian $W_c$ and observability Gramian $W_o$ are expressed as infinite integrals [66]:

$$W_c = \int_0^\infty e^{At} BB^T e^{A^T t} dt,$$

$$W_{o,\ j \in Y} = \int_0^\infty e^{A^T t} C_j^T C_j e^{At} dt \tag{19}$$

$W_c$ captures how easily each internal state is reached by the inputs [67] and $W_{o,j}$ how each internal state affects each output $j$ (j is part of $Y$) [68]. They are solutions to the Lyapunov equations:

$$W_c = \int_0^\infty e^{At} BB^T e^{A^T t} dt,$$

$$W_{o,\ j \in Y} = \int_0^\infty e^{A^T t} C_j^T C_j e^{At} dt \tag{20}$$

The product between the two Gramians gives the Hankel matrix $H$ [69]

$$H = W_c W_{o,j \in Y} \tag{21}$$

which reflects each state's combined observability and controllability, i.e., how much each state contributes to outputs but also how easily it is controlled by the inputs. The Hankel singular values (HSVs) $\sigma_i$ [70,71] are defined as

$$\sigma_i = \sqrt{|\lambda_i|} \tag{22}$$

where $\lambda_i$ are the eigenvalues of $H$, quantify the energy of each state or how important it is in terms of dynamics. HSVs decay rapidly in large-scale systems [72], which facilitates low-rank approximations by taking threshold values for truncation at the point of decay. The features in the multiome data corresponding to large HSVs were kept and system reidentification for all linear matrices ($A$, $B$ and $C$) was carried out in the reduced subspace using the optimal parameters chosen by the hyperparameter search of the multi-objective function (12).

**Koopman matrix alignment between species**

Alignment of the new reduced Koopman matrices was carried out before applying control by using a transformation matrix $T$ to align the Koopman matrix of *C. major* ($A_1$) to that of *C. rosea* ($A_2$) as in

$$\min_T \|C_2 A_2 - C_1 A_1 T\|_{fro} \tag{23}$$

Transformation matrix $T$ aligns both Koopman matrix $A_1$ to $A_2$ akin to Procrustes analysis of vector fields [73] but each matrix is also multiplied by their respective output matrices $C$ which means $T$ jointly minimizes the distance between internal and output dynamics of the two systems. This allows the mapping of the internal state of one system to match that of another while at the same time being projected onto their respective outputs. The new Koopman matrix for *C. major*

 

$$\widetilde{A_1} = A_1 T \tag{24}$$

is used as input to linear MPC. The Koopman operator of *C. major* has to firstly be projected in the right subspace (by transformation matrix *T*) for correct alignment of the product of Koopman and output matrices between the two species before optimal control can find the best control input solutions for proper phenocopying.

**Linear model predictive control**

MPC is implemented as in [74], where *C. major* inputs are optimized to match the outputs of *C. rosea* for correct phenocopying. The cost function is defined as

$$J = \sum_{i=1}^{t} \left[ \left(z_i - z_{ref}\right)^T Q \left(z_i - z_{ref}\right) + \left(u_i^T R_1 u_i\right) + \left(x_i^T R_2 x_i\right) \right] \tag{25}$$

where *z* are the outputs of *C. major*, $z_{ref}$ the outputs of *C. rosea*, $Q$, $R_1$ and $R_2$ the diagonal output, control input and input penalty matrices respectively, and *t* the number of timepoints over which *to* iterate, with a one-timestep prediction at each iteration (assuming perfect outcome). The objective is to minimize the cost function *J* by finding optimal inputs $u_i$ to minimize the difference between the outputs of the two systems. The input-output equations are placed as constraints to the objective function as in:

$$z = Cx$$
$$x_{i+1} = \widetilde{A_1} x_i + B_1 x_i$$
$$u_i = -K \left(x_i - x_{ref}\right) \tag{26}$$

where *K* is a gain matrix to be found, *x* the *C. major* inputs to be optimized, $x_{ref}$ the known *C. rosea* inputs, $\widetilde{A_1}$ the Koopman operator, *C* the output operator and *B* the control operator. The effectiveness of input control is evaluated based on reconstruction accuracies ($R^2$) of output values of *C. rosea*. The gain matrix *K* was found by solving the discrete-time algebraic Riccati equation [75]

$$\widetilde{A_1}^T P \widetilde{A_1} - P - \left(\widetilde{A_1}^T PB_1\right) \left(R_1 + B_1^T PB_1\right)^{-1} \left(B_1^T P \widetilde{A_1}\right) + R_2 = 0 \tag{27}$$

to find P with $R_1$, $R_2$ and Q are the weight matrices used in (25) and are vital for optimizing the control inputs and minimizing cost *J*. $R_1$ and $R_2$ are identity matrices and Q has a weight of $5^{10}$ in its diagonals putting higher weight on matching outputs. *P* is then used as input to

$$K = \left(R_1 + B^T PB\right)^{-1} B^T P \widetilde{A_1} \tag{28}$$

to find *K*.

**Sensitivity gain matrix**

The importance of each feature in linear MPC was evaluated based on the sensitivity gain matrix [76] *S* found by

$$S = \left(\widetilde{A_1} - BK\right)^{-1} B \tag{29}$$

of which the mean for every feature was taken after many iterations. Sensitivity values give a measure of how small changes in the input influence the state dynamics. The full derivation of *S* (29) can be found in S1 Appendix.

## Code implementation

We use the *cvxpy* package in Python for the multi-objective optimizations with the 'CLARABEL' solver as it produced the highest fit accuracy scores for our quadratic objective functions. We parallelize the code using *loky* backend to iterate the algorithm 100 times, reducing computation time from hours to a minute. Multiple iterations were required to validate robustness of results across repetitions as well as the calculation of consensus eigenmodes and mean sensitivity values. A leave-3-out cross validation was repeated 10 times to validate the main model, by excluding the last 3 timepoints in the test set. The following versions of python and packages were used to create a virtual environment: *python* 3.13.7, *pandas* 2.3.3, *numpy* 2.2.6, *scipy* 1.16.2, *cvxpy* 1.6.0, *scikit-learn* 1.7.2, *joblib* 1.5.2, *matplotlib* 3.9.4, *networkx* 2.5, *seaborn* 0.13.2.

## Results

### Part I: System identification

**Kernel choice determines the quality of nonlinear multiome integration.** We evaluated four kernel families (linear, Gaussian, polynomial, and sigmoidal) within the multi-objective optimization framework to determine which best captured the nonlinear dynamics of the concatenated transcriptome, proteome, and metabolome datasets using kernel DMD (see Methods). For the data and each output, we evaluated the reconstruction accuracy ($R^2$ score) using the full Koopman operator in linear space which was derived by the using the learned weights of the kernel in (12) and the kernel gradient (14). The kernel evaluations and the hyperparameter search results are shown in Table 2. The accuracy of the sigmoidal kernels remains consistently high (around 0.7) for both species regardless of the coefficient of the sigmoid (relative to the linear decay) term. The accuracy decreases when the coefficient $\gamma$ increases combined with a higher sigmoid kernel term. As the term $\gamma$ represents the interaction term between inputs, when it is larger it tends to overfit noise and outliers and does not generalize over the whole input space. It thus creates a steeper threshold boundary that might create vanishing gradients used for the linearization step (13, 14) that does not fall within the base state (mean) of the inputs. The robustness of the sigmoidal kernel was also very high with <0.05 standard deviations of $R^2$ score. Comparing with the other kernels, the sigmoidal kernel performed much better than the gaussian or polynomial kernels, justifying its choice for fitting to this data. Where usually the gaussian kernel almost always fits well to any dataset due to its good generalizability [37,77,78], here it gave much lower accuracies. It should be noted that the success of the sigmoidal kernel is attributed to its second linear degradation part $x^T x$ which is also used on its own as part of the linear kernel (hence the high accuracy

**Table 2. Kernel function evaluations.**

| kernel | Hyperparameters | | | | | Reconstructions ($R^2$ score) | |
|---|---|---|---|---|---|---|---|
| | $\sigma$ | $d$ | $\gamma$ | $c$ | $c_s,c_l$ | *C. major* (control) | *C. rosea* (stress) |
| Linear | | | | | | 0.73 +/- 0.03 | 0.64 +/- 0.02 |
| Gaussian | 0.0001 | | | | | 0.05 +/- 0.0 | 0.05 +/- 0.0 |
| | 0.001 | | | | | -1.12 +/- 0.04 | -0.88 +/- 0.07 |
| Polynomial | | 3 | 0.0001 | 1 | | 0.05 +/- 0.0 | 0.05 +/- 0.0 |
| | | 5 | 0.0001 | 1 | | 0.05 +/- 0.0 | 0.05 +/- 0.0 |
| | | 3 | 0.001 | 1 | | 0.05 +/- 0.0 | 0.05 +/- 0.0 |
| Sigmoidal | | | 0.001 | -0.0002 | 3, 0.1 | 0.75 +/- 0.01 | 0.71 +/- 0.02 |
| | | | 0.001 | -0.0002 | 0.1, 0.1 | 0.76 +/- 0.02 | 0.68 +/- 0.04 |
| | | | 0.001 | -0.0002 | 5, 0.1 | 0.76 +/- 0.03 | 0.67 +/- 0.04 |
| | | | 0.1 | -0.0002 | 3, 0.1 | -1.89 +/- 0.0 | -26.40 +/- 0.0 |

of the latter). If this term is added to any other kernel along with its corresponding derivative term in the gradient, then the linear model of the respective kernel succeeds in reconstructing the data and outputs with high accuracies regardless of the first part of the kernel (whether it be gaussian, polynomial or sigmoidal). We choose to proceed with the "Hill-function"-like kernel to showcase that tailored kernels can be combined to produce representations of functions applied to molecular systems and lead to conclusions aligned with the literature in relation to distinctions between C3 and CAM photosynthesis as shall be presented later in the results. Each kernel was also evaluated based on the ability of the linear Koopman operator and its eigenmodes to reconstruct each output $Y$ (12). Output reconstructions of the sigmoidal kernel using the linear Koopman operator were moderate to high (>0.6 $R^2$ Fig 3) for all outputs and both species. To avoid risk of over-fitting due to high-dimensionality of the data, we repeated a leave-3-out cross validation 10 times by excluding the last 3 timepoints in the test set. Overall $R^2$ accuracy in the test set was high (0.7 for data, average 0.7 for outputs), which indicated that the model trained on the training data generalized well and did not overfit.

**Eigenmode decomposition reveals latent temporal modules.** The eigendecomposition of the linear Koopman operator (see Methods) results in the decomposition of dynamics into distinct temporal components, or eigenmodes, each representing a dynamical module contributing to the evolution of the system over the diurnal cycle. Each eigenmode and its temporal dynamic (Fig 2) can be linked to each output via reconstruction accuracy ($R^2$ score; Fig 3). Modal reconstructions provide an indirect link between features (through the rank of the feature's absolute value in the eigenmode) and how they best predict each output across the timeseries. Thus, a direct comparison can be made between the underlying multiome pathways that differentially predict "C3" and strong CAM outputs of the two species. Figs 2 and 3 show the eigenvalues, eigenmode amplitudes, and mode-specific reconstruction accuracies from one representative iteration of the kernel-DMD optimization. Because kernel-based system identification involves stochastic initialization and resampling, individual eigenmodes vary slightly between iterations. Therefore, these figures serve to illustrate the spectral structure and qualitative modal types (stable, decaying, oscillatory). To extract biologically meaningful information, we do not interpret individual modes from a single iteration. Instead, we compute a consensus across 100 iterations, retaining only those eigenmodes that consistently predict $CO_2$ gas-exchange dynamics ($R^2 > 0.3$). The consensus eigenmode topologies (Fig 4 and S1–S2 Tables) provide the stable spatial patterns that persist despite stochastic variability and are biologically interpretable. These are the modes used in all pathway-level and biological interpretations discussed below. Importantly, the eigenmodes emerging from the Koopman operator naturally contain a mixture of transcript, protein, and metabolite features. This occurs because the kernel and graph Laplacian couple features across omics layers, enabling the model to learn their coordinated temporal behavior rather than treating each layer independently. The resulting consensus eigenmode clusters (Fig 4) therefore represent functional modules spanning transcriptional regulation, enzymatic activity, and metabolic flux, consistent with the cross-omic coordination required for CAM and C3 photosynthesis.

**Consensus eigenmode topology identifies stable cross-omic modules.** The C. rosea consensus network (16) shows a strongly connected module enriched for glycolysis and sugar-mobilization pathways, including key enzymes such as PFK3, FBA2, GAPA, and GAPC as well as metabolites such as glucose, fructose, sucrose, mellibiose, trehalose, myo-inositol and raffinose (Fig 4A and S2 Table). Multiple nodes associated with starch breakdown and sucrose cycling (AGPP, PHO1, RFS1, BGAL1) appear as high-rank hubs, suggesting coordinated remobilization of carbohydrate pools during the night. Entry into glycolysis starts from carbon breakdown where Glucose-1-phosphate Adenylyltransferase (AGPP), a small subunit of the ADP-glucose pyrophosphorylase (AGPase) enzyme complex, catalyzes the conversion of glucose-1-phosphate to form ADP-glucose [79] a precursor for starch synthesis. Following carbon breakdown of starch to maltose and then glucose by various intermediate enzymes such as phosphoglucan water dikinase (GWD3) or alpha-glucan phosphorylase (PHO1), glucose-6P enters the glycolysis pathway. Glucose-6P is then converted to fructose-6P which is converted to sucrose-6P and eventually sucrose. *C. rosea* was found to switch from glucose and fructose sugar pools to sucrose accumulation for osmoprotection and energy reservation [80] during low photosynthetic

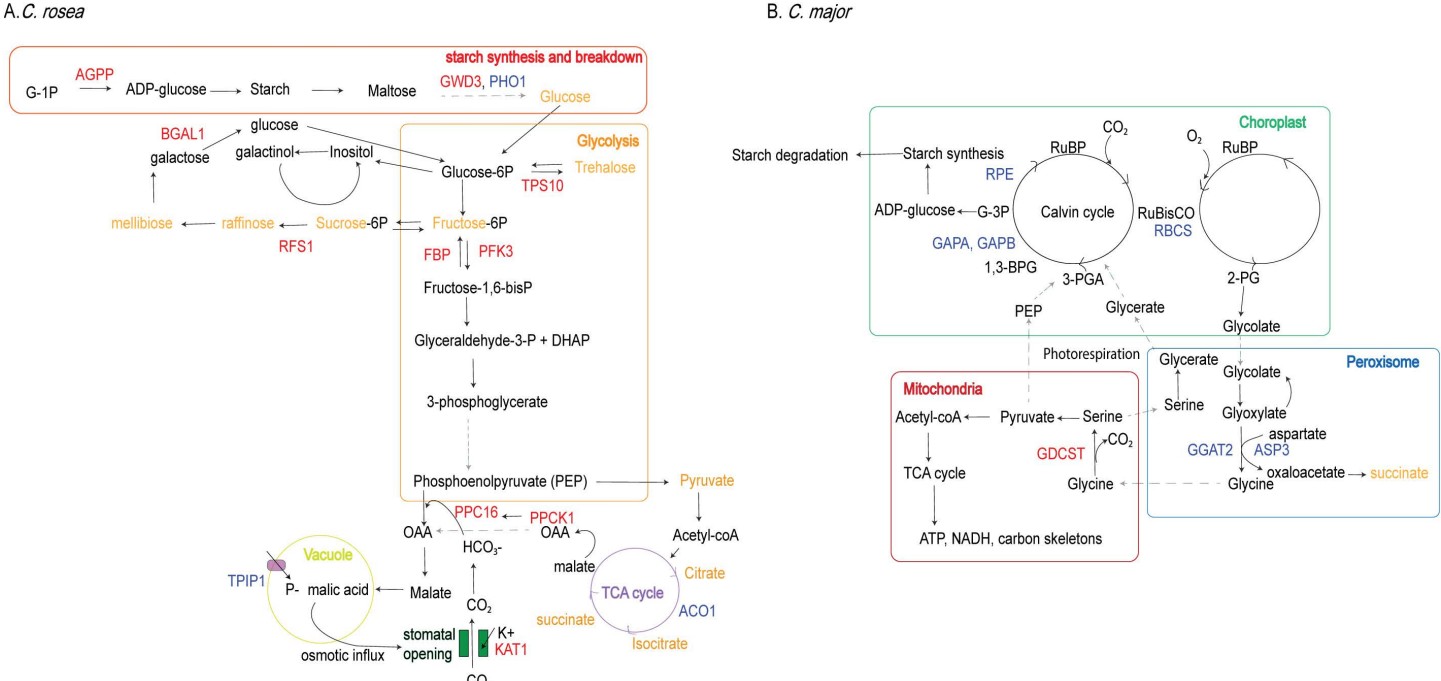

**Fig 4. Consensus eigenmode topology for each species.** The networks shown represent the consensus topology of eigenmodes that consistently predicted $CO_2$ gas-exchange outputs across 100 kernel-DMD iterations ($R^2 > 0.3$). Nodes correspond to multiome features (transcripts, proteins, metabolites), coloured according to whether they are transcript [12], protein (blue) or metabolite (orange). Edges represent feature pairs that co-occurred within the same eigenmode with frequency >0.5 across iterations, reflecting stable dynamical associations. **(A)** *C. rosea*. The consensus clusters highlight enhanced flux through glycolysis, starch remobilization, and key CAM-associated enzymes (e.g., PEPC, PPCK1, PHO1). Features from the TCA cycle (ACO, CAC3) and sugar interconversion pathways (RFS1, BGAL1) also appear as highly ranked and strongly connected nodes. This topology reflects a night-time metabolic program supporting PEP regeneration, organic-acid accumulation, stomatal opening at night, and CAM-type carbon assimilation. Together, these consensus graphs reveal stable dynamical modules that differentiate C3 (*C. major*) and strong CAM (*C. rosea*) metabolism, independently of stochastic variation across individual DMD iterations. **(B)** *C. major*. The dominant consensus cluster is enriched for photorespiratory enzymes (e.g., RBCS, GDCSH, GDCSP, CTIMC), Calvin-cycle intermediates, and glyoxylate metabolism. This topology indicates that *C. major* dynamics are strongly shaped by daytime RuBisCO oxygenase activity, carbon recycling through the photorespiratory pathway, and metabolic routing toward 3-PGA production.

activity diminished under stress, sustaining nocturnal CAM metabolism. Other sugar interconversions, such as inositol and galactose interconversions, that link to glycolysis enhance accumulation of PEP during the night. Galactinol--sucrose galactosyltransferase 1 (RFS1) which was found within the consensus eigenmode in *C. rosea*, catalyzes the conversion of sucrose to raffinose via the interconversion of galactinol to inositol, eventually leading to melibiose synthesis. Galactose, a product of melibiose, is converted back to glucose by beta-galactosidase (BGAL1) and glucose changes to glucose-6P which again enters glycolysis. Small sugars are thought to act as osmolytes within guard cells of stomata to increase turgor and cause stomatal opening or alternatively act as energy reserves for respiration [81]. Fructose-6-P (coming from Glucose-6P) leads to fructose-1,6-bisP using Phosphofructokinase 3 (PFK3), which then forms glyceraldehyde-3-P. This is then converted to 3-phosphoglycerate and finally produces PEP in the cytosol. PEP then enters carboxylation pathways where phosphoenolpyruvate carboxylase (PEPC) converts it to oxaloacetate which then forms malate by malate dehydrogenase. PEPC is regulated by phosphoenolpyruvate carboxylase. Therefore, increased flux through starch remobilization and glycolysis leads to more production of PEP that is ready to be an acceptor to more bicarbonate ions for production of malic acid, enforcing stomatal opening at night for $CO_2$ assimilation. In turn malic acid accumulation leads to osmotic influx in the vacuole, further promoting guard cell turgor and stomatal opening. Evidence of elevated PEP

production comes also by presence of TCA cycle intermediates, since PEP is a precursor of pyruvate which is converted to acetyl-coA that is the entry point to the TCA cycle. Citrate is converted to isocitrate by aconitase (ACO) within the cycle. TCA cycle by-products and intermediates like citric acid, oxaloacetate or malate result in higher availability and storage of organic acids in the vacuole that then elevate turgor pressure in guard cells resulting in stomatal opening. Furthermore, interconversions with pyruvate and PEP provide more substrate availability for $CO_2$ assimilation promoting stomatal opening. Importantly, CAM hallmark enzymes PEPC and PPCK1 also emerge as strongly ranked and densely connected nodes, reinforcing their central role in night-time malate accumulation. Together, these modules reflect a metabolic program optimized for night-time stomatal opening, organic-acid accumulation, and PEP availability, all characteristic of CAM photosynthesis.

In contrast, the *C. major* consensus topology is dominated by enzymes linked to photorespiration (RBCS, GDCST, GGAT2, ASP3) and Calvin-cycle regeneration (GAPA/B, RPE) (Fig 4B and S1 Table). This indicates that RuBisCO oxygenase activity and its associated carbon-recycling pathways play a major role in shaping *C. major* dynamics. The presence of these photorespiratory components in the consensus structure suggests that carbon flow is preferentially routed toward daytime $CO_2$ assimilation and recycling, rather than toward the night-time accumulation of organic acids. Because photorespiration consumes energy and reduces the availability of PEP for nocturnal carboxylation, this topology aligns with the observed C3 phenotype of *C. major*—namely, limited night-time stomatal opening and weaker CAM flux compared to *C. rosea*. During daytime and with open stomata in *C. major* high oxygen concentrations lead to oxygenase activity of RuBisCO to produce 2-Phosphoglycolate (2-PG). Since 2-PG is toxic and cannot enter the Calvin cycle directly, it is processed through photorespiration by first being converted to glycolate by 2-Phosphoglycolate Phosphatase. It then enters the peroxisome and is converted to glyoxylate. Glyoxylate interconverts with glycolate or goes on to form glycine by Glutamate--glyoxylate aminotransferase (GGAT2) and is then transported to the mitochondrion. There, it reacts to produce serine by an amino-methyltransferase (GDCST), which can then follow two routes: it either goes back in the peroxisome or is converted to pyruvate with downstream products in the TCA cycle. Serine transported back in the peroxisome is converted to glycerate and then 3-PGA which can now enter the Calvin cycle back in the chloroplast. With carbon flowing towards the Calvin cycle (GAPA, GAPB, RPE) during the day, *C. major* has less substrate available for $CO_2$ assimilation at night which doesn't promote its stomata opening at that time. It should also be noted that in the *C. rosea* consensus eigenmode there were also photorespiratory biomarkers (GDCSH, GDSCP, GLYR; see S2 Table) which points to RuBisCO maintenance despite closing of stomata as also evidenced in *Guzmania monostachia* during a C3-CAM transition [82]. This is probably to the high decarboxylation activity during the day that replenishes $CO_2$ near RuBisCO's active site.

## Part II: System control and phenocopy of CAM dynamics

**Control directions derived from nonlinear kernel residuals.** For the identification of optimal control inputs that can shift the output dynamics of *C. major* to those of *C. rosea* and effectively achieve the phenocopy of strong CAM in a "C3" molecular background, we needed to find an appropriate control matrix $B$ [83] that could direct control inputs to the right internal system states $x$ (the concatenated multiome data). In this study, 'control' refers to model-driven perturbations of internal molecular states—transcripts, proteins, or metabolites—that would redirect the system's trajectory toward CAM-like outputs. The control matrix $B$ identifies directions in the multiome space that act as effective levers, such that small changes along these directions yield large changes in the system's dynamical behavior. Biologically, these correspond to hypothetical manipulations of gene expression, enzyme abundance, or metabolite pools. Within the kernel-DMD framework, learning appropriate control laws [37] requires already identified external inputs $u$ which are assumed to exhibit nonlinear pairings with the internal state variables $x$. In our case, we did not have appropriate measurements available that could act as external control inputs and we limited the system to be autonomous with internal feedback control loops emerging within the interconnectivity of the multiome itself. The identification of suitable control mechanisms

was therefore derived using only information coming from internal system dynamics. One of the major contributions of this paper is the utilization of the nonlinear residual parts ($N$) of the Taylor series expansion (13) as nonlinear forcing constraints. These higher order terms were briefly mentioned by [37] but were never explicitly used for subsequent steps in relation to control. Here, we make use of $N$ to derive the control matrix $B$ (see Methods section) and the nonlinear inputs $u$, with the assumption that it consists of higher order terms that can act as nonlinear input constraints to the linear Koopman operator (14) and is sensitive to external parameters that can influence and drive the internal dynamics of the multiome system.

**Dynamic feature selection via Hankel singular values.** Due to the high dimensionality of the system, a smaller state space representation of dynamics was sought using dynamic information coming from the Koopman ($A$) and control input ($B$) matrices. Feature selection was carried out according to observability and controllability metrics, a method largely developed by Willems [84] and later used by others [85]. Model reduction was carried out by taking the first 50 indices of HSV values after ranking them in descending order for all outputs. The number of indices was selected as a balance between good reconstruction accuracy of *C. rosea* outputs after linear Model Predictive Control (MPC, see Methods) in the selected reduced feature space of *C. major* and minimal number of features kept. The final reduced features spaces included unique features across both *C. rosea* and *C. major* according to this ranking for each output. The top features selected over 100 iterations include glycolysis enzymes such as GAPC, FBA, GAPA and FBP as well as Calvin cycle intermediates such as glycine dehydrogenase (GDCSP) and transcription factors GATA1 [86] and PCF1 [87], involved in regulating gene expression in response to nutrient availability and light intensity, helping regulate stomatal movement and stress responses. The robustness of HSV rank as a feature selection method is shown in Fig 5A which shows the features retained after data reduction and their frequency over 100 iterations. Thus, a solid observability-controllability metric (the HSV rank) was used to select features which play dual roles: they strongly shape system outputs and their levels can be effectively influenced under control. This data reduction step leads to efficient application of control in a smaller subspace.

**Phenocopying CAM in *C. major* using model predictive control (MPC).** Using the aligned Koopman operators (after transformation using projection matrix $T$, see Methods) and reduced state space, we applied linear MPC to drive the *C. major* system toward *C. rosea* trajectories. The controlled system reproduced the majority of CAM outputs with moderate to high fidelity (mean $R^2 = 0.38$–$0.9$ across outputs, Fig 5B). Thus, despite the underlying nonlinearity, a linear control law is sufficient when derived from the Koopman representation. Predictive accuracy quantifies how well the controlled *C. major* system reproduces the experimentally measured diurnal trajectories of *C. rosea* outputs (malate, $CO_2$ exchange, PEPC, PCK1, etc.). Thus, an $R^2$ value indicates the degree to which the modeled phenocopy matches the true CAM phenotype, rather than experimental validation of the inferred control inputs. It should be noted that higher mean accuracies of the phenocopy were observed when during system reidentification the inputs of *C. rosea* under control were mapped to the *C. rosea* outputs under stress for identification of the output matrix C in the reduced system of *C. rosea*. If both inputs and outputs were taken from the same condition, phenocopy accuracies were reduced (mean $R^2 = 0.2$–$0.45$ across outputs). This could be because the inputs of *C. rosea* under control might be closer in overall distribution to those of *C. major* under control, so easier to shift, while the outputs of *C. rosea* under stress provide a more prominent form of CAM.

**Sensitivity gain analysis identifies regulatory levers of CAM induction.** Sensitivity values derived from the control gain matrix (29) highlight features whose modulation most strongly affects controlled outputs. According to sensitivity values, the most important features are shown in Fig 5C with positive values indicating entries which are positively affected by control, negative entries involved in negative feedback loops and values near 0 where internal states are not influenced a lot by control. Of note is that all of the entries are transcripts which were selected as top dynamic regulators according to the HSV data reduction step and were also found to be important as part of regulation. Of the top positive values, some (PFK3, GAPN, FBA2 and SPS1) belong to sugar mobilization and glycolysis that was found as a key pathway in the consensus eigenmode in *C. rosea* which signifies its importance in regulating CAM in this species.

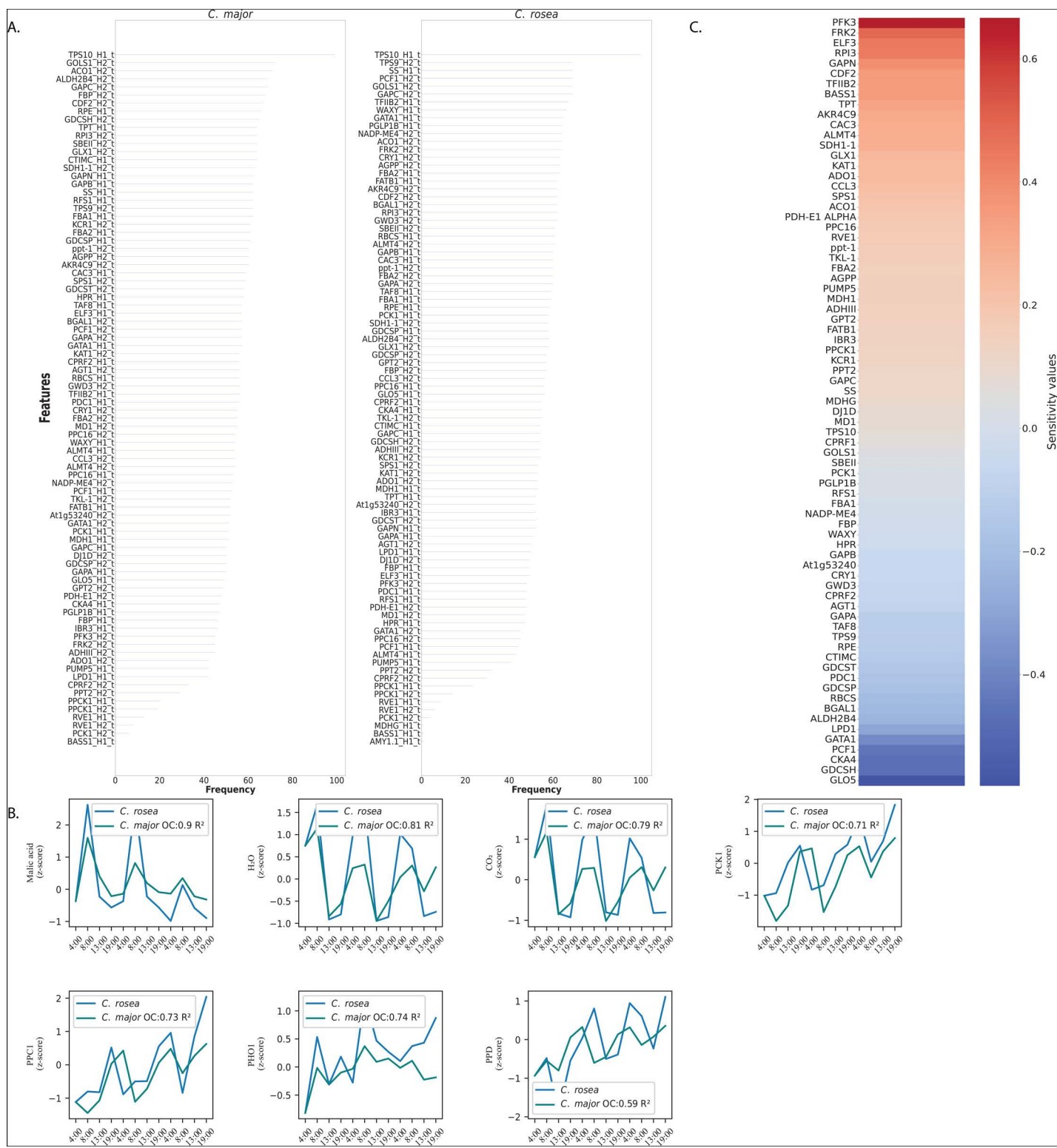

**Fig 5. System Control output results. A)** Robustness of data reduction via HSV feature selection across iterations. Data reduction was carried out by truncating Hankel Singular Values below a threshold. The frequency of each retained feature after 100 iterations is shown on the x-axis for *C. major* and *C. rosea*. **B)** Phenocopy of the *C. rosea* outputs in *C. major*. Prediction accuracy (*R²* score) *C. rosea* outputs in *C. major* after MPC optimization of

control inputs. Relatively high accuracies are achieved after implementing the control strategy to shift the *C. major* output distributions to match those of *C. rosea*, highlighting the success of the workflow. **C)** Sensitivity gain matrix values. Positive entries [12] indicate features with higher positive feedback input in *C. major* and negative entries (blue) features with negative feedback input. Glycolytic enzymes (PFK3, GAPN, FBA2 and SPS1) mostly have positive feedback and photorespiratory enzymes (RBCS, GDCSH, GDCSP) have negative feedback. This agrees with the widely known role of glycolysis in maintaining CAM levels and of photorespiration being active in C3 types of photosynthesis.

Negative sensitivities are obtained for photorespiratory enzymes (RBCS, GDCSH, GDCSP). The sensitivity profile thus mirrors the eigenmode results of Part I (Fig 4), suggesting that activation of glycolysis/TCA flux and suppression of photorespiration form the mechanistic basis of the C3-to-CAM transition.

## Discussion

We developed a workflow for short timeseries multiome data integration that is deeply embedded within dynamical systems theory using kernel DMD. In the first part of our pipeline, we carry out System Identification. The integration of the transcriptome, proteome and metabolome datasets is not purely statistical, although they are concatenated into a single multiome state vector. Functional relationships among omics layers are encoded through the use of a kernel, which computes similarity measures between all feature pairs, allowing transcripts, enzymes, and metabolites to jointly influence the learned dynamics in the induced Hilbert space. In addition, incorporation of the graph Laplacian (see Methods) embeds prior biological connectivity into the sigmoidal kernel. This ensures that molecular relationships such as transcript→protein→metabolite flux are represented in the model. The resulting Koopman operator captures the joint temporal evolution of the multiome system, producing eigenmodes that contain mixed omics information reflecting their coupled dynamical behavior. Consensus eigenmodes reveal distinct dynamical architectures underlying CAM and C3 photosynthesis. *C. rosea* exhibits coordinated nocturnal flux through glycolysis, TCA-cycle intermediates, and starch–sucrose interconversion pathways—supporting PEP regeneration and malate accumulation central to CAM. In contrast, *C. major* shows strong photorespiration and Calvin-cycle regeneration, consistent with predominantly daytime $CO_2$ fixation and reduced CAM activity.

In the second part of our pipeline we carry out System Control. In our context, "control" does not represent experimentally applied perturbations but rather model-derived directions in the multiomic state space that predictably alter system trajectories. Control inputs should be interpreted as hypothetical molecular manipulations—changes in transcript levels, enzyme activities, or metabolite pools—that guide the system toward desired outputs (here, CAM-like behavior). This theoretical formulation offers mechanistic hypotheses for future experimental testing. Phenocopy accuracy reflects how well the controlled *C. major* system reproduces the temporal profiles of measured CAM outputs in *C. rosea*. This measure does not imply direct biological validation of control inputs; instead, it verifies that the inferred dynamical structure supports CAM-like trajectories when perturbed along specific directions.

Various glycolytic enzyme transcripts were found to be positively activated in order for successful phenocopy of the CAM signature to occur. Of those, Phosphofructokinase (PFK3) is converts fructose-6P to fructose-1,6-bisP and is involved in the downstream accumulation of malate. During transition from C3 to CAM, this could be a key regulator for shifting night-time accumulation of PEP and constraining stomatal opening during the night [88]. Non-Phosphorylating Glyceraldehyde-3-Phosphate Dehydrogenase (GAPN) oxidizes glyceraldehyde-3-P to 3-phosphoglycerate as part of glycolysis. The role of GAPN in the transition from C3 to CAM has been investigated before by [89] who showed that a NAD-glyceraldehyde-3-phosphate dehydrogenase (NAD-GAPDH) had an increased transcript expression in transition from C3 to CAM in *M. crystallinum*. Fructose-Bisphosphate Aldolase (FBA2) catalyses the forward reaction of fructose-1,6-bisP to dihydroxyacetone phosphate (DHAP) and glyceraldehyde-3P (G-3P) during glycolysis and showed increased expression in the induction of C3 to CAM after salt treatment in *M. crystallinum* [90]. In another study for the C3 to CAM transition in *M. crystallinum*, FBA was found to correlate with leaf microarray data along with PEPC, an inositol-3-P synthase and RuBisCO small chain and were differentially regulated [91]. Sucrose-Phosphate Synthase (SPS1) is involved

in sucrose synthesis as part of the starch and sucrose metabolism and converts UDP-glucose and fructose-6P (part of glycolysis pathway) to sucrose-6P which is then further converted to sucrose. SPS was shown to be positively correlating with drought and waterlogging stress [92] and controls the partition between starch and sucrose levels [93]. Sucrose, along with glucose and fructose, has been found to be associated with malate synthesis in *Clusia* species during the night [94]. TCA cycle enzymes like Succinate dehydrogenase (SDH1–1) and Citrate synthase (CAC3) also had positive values $S$ highlighting the possible involvement of the TCA cycle in providing more flux towards oxaloacetate for increasing $CO_2$ assimilation during the night. Increase in glycolytic flux and flux through the TCA cycle (which was modelled as the succinate dehydrogenase flux) during the night was also predicted by metabolic modelling of the C3-CAM continuum [95]. Transcription Factor IIB (TFIIB) forms part of the regulation of the shift and could be involved in the regulation of stomatal opening as well as control of genes involved in carbon fixation. It is a widely functionally diversified transcription factor across the plant kingdom [96]. The obvious role of malate dehydrogenase (MDH1) and phosphoenolpyruvate carboxylase (PPC16) places them as good candidates for participating in the C3-CAM switch (Fig 5C).

Negative feedback appears to be in place for features involved in photorespiration and the Calvin cycle such as Glycine Decarboxylase Subunit H (GDCSH), Ribulose-1,5-bisphosphate Carboxylase/Oxygenase (RBCS), Cytosolic Triose Phosphate Isomerase-like protein (CTIMC) and Glycine Decarboxylase Subunit P (GDCSP) (Fig 5C). As mentioned above, photorespiration is downregulated during CAM [97] due to its expense in energy conservation and the $CO_2$ concentrating mechanism around RuBisCO during the day in CAM plants. It therefore makes sense that enzymes involved in photorespiration are negatively regulated. Therefore, the control method implemented here is congruent with the eigenmode analysis that differentiates strong CAM and C3 by glycolysis, TCA cycle and photorespiratory pathways and is also in agreement with the related literature. The control predictions presented are theoretical and serve to generate mechanistic hypotheses about molecular drivers of C3-to-CAM transitions. Although not validated experimentally within this study, the model identifies regulators consistent with CAM biology and offers specific targets for future experimental perturbation.

In conclusion, we used a parameter-free data-driven approach for the integration of short time series data in the multiome setting. After evaluating different linear and non-linear kernels, we concluded that the linear part of the sigmoidal kernel best fits our data and could be appropriate for other multiome data as well. By combining the use of eigen decompositions, data reduction and control we managed to identify key dynamic and topological differences between the two time-resolved multiome networks to better distinguish the two types of photosynthesis (C3-like and strong CAM). This happens mainly through increased activity in glycolysis, sugar intermediates and the TCA cycle. Through our control strategy, we identified key control biomarkers that enable the phenocopying of CAM phenotypes within a C3-physio background. In follow-up studies more methods such as hierachical kernels for different omic layers and the use of multimodal Koopman autoencoders to integrate multiome data will be tested. It is crucial though to consider the dimensionality of the time series data as well which, for our type, a multimodal autoencoder would likely overfit. We therefore conclude that this workflow is suitable for such types of data and results in a plethora of conclusions that provide functional information which confirms and complements previous established knowledge on CAM photosynthesis and plasticity. The framework could be extended through experimental perturbation of key regulators (PFK3, FBA2, GAPN, SPS1), integration with metabolic flux models to resolve pathway-level changes and application to other environmental stress responses across plant systems and other contexts such as biomedical applications. Overall, the integration of kernel-based system identification and Koopman control provides a flexible platform for dissecting complex, multi-layered biological dynamics and generating testable hypotheses about regulatory mechanisms.

## Supporting information

**S1 Table. Consensus eigenmode topology *C. major*.** Edge table that includes node feature names and edge weights for a selected eigenmode of *C. major* discussed in the manuscript.
(CSV)

**S2 Table. Consensus eigenmode topology *C. rosea*.** Edge table that includes node feature names and edge weights for a selected eigenmode of *C. rosea* discussed in the manuscript.
(CSV)

**S1 Fig. Main pathway flows in C3 and CAM photosynthesis.** A) In C3 photosynthesis $CO_2$ is assimilated in the Calvin cycle in the morning where it is converted to 3-phosphoglycerate during primary carbon fixation. The hydrolytic route is the main pathway for carbon breakdown. B) In CAM photosynthesis, $CO_2$ assimilation occurs at night to reduce water loss during the day by restricting stomatal opening in the night. Carbonic Anhydrase (CA) convers $CO_2$ into hydrogen bicarbonate ($HCO_3^-$) that then reacts with PEP to produce oxaloacetate (OAA) catalysed by PEPC. OAA is converted to malate that is stored as malic acid in the vacuole during the night. During the day it is converted back to malate that is decarboxylated by PEPCK into PEP or enters gluconeogenesis that then leads to conversion of PEP to starch. During the night starch breakdown occurs mainly via the phosphorolytic pathway that regenerates PEP to accept more $CO_2$.
(EPS)

**S1 Appendix. Supplementary information for parts of the mathematical derivations.** Specifically, the analogy between the sigmoidal kernel and Hill function dynamics is made clearer along with the derivation of the sensitivity gain matrix (29).
(DOCX)

## Acknowledgments

The authors acknowledge the use of ChatGPT for assistance with English language correction during the review process. However, all other aspects of the paper, including the research, analysis, and conclusions, were conducted and validated solely by the authors.

## Author contributions

**Conceptualization:** Iro Pierides, Wolfram Weckwerth.

**Data curation:** Hannes M. Kramml.

**Formal analysis:** Iro Pierides.

**Funding acquisition:** Wolfram Weckwerth.

**Methodology:** Iro Pierides, Steffen Waldherr, Wolfram Weckwerth.

**Resources:** Wolfram Weckwerth.

**Supervision:** Steffen Waldherr, Wolfram Weckwerth.

**Validation:** Steffen Waldherr.

**Visualization:** Iro Pierides.

**Writing – original draft:** Iro Pierides.

**Writing – review & editing:** Iro Pierides, Hannes M. Kramml, Steffen Waldherr, Wolfram Weckwerth.

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
