## [Decision Letter · Decision Letter 0]

21 Nov 2025

Kernel-DMD with control for phenocopying of complex biological systems

PLOS Computational Biology

Dear Dr. Pierides,

Thank you for submitting your manuscript to PLOS Computational Biology. After careful consideration, we feel that it has merit but does not fully meet PLOS Computational Biology's publication criteria as it currently stands. Therefore, we invite you to submit a revised version of the manuscript that addresses the points raised during the review process.

We look forward to receiving your revised manuscript.

Kind regards,

Julio R. Banga, Ph.D.

Academic Editor

PLOS Computational Biology

Pedro Mendes

Section Editor

PLOS Computational Biology

**Additional Editor Comments:**

Following the evaluation of your manuscript by four reviewers, it has been determined that the paper requires major revision before it can be considered for publication. While the reviewers acknowledge the potential significance of your study, they have raised substantial concerns regarding the clarity of the methodology and the depth of the discussion in relation to existing literature. In addition, several sections would benefit from improved organization and more precise language to enhance readability. We kindly ask that you address each of the reviewers’ comments in detail, providing clear explanations of the changes made or justifications where revisions are not possible. A carefully revised version that responds comprehensively to these points will allow us to reassess the manuscript for possible acceptance.

**Journal Requirements:**

At this stage, the following Authors/Authors require contributions: Iro Pierides, Hannes M. Kramml, Steffen Waldherr, and Wolfram Weckwerth. Please ensure that the full contributions of each author are acknowledged in the "Add/Edit/Remove Authors" section of our submission form.

3) Some material included in your submission may be copyrighted. According to PLOSu2019s copyright policy, authors who use figures or other material (e.g., graphics, clipart, maps) from another author or copyright holder must demonstrate or obtain permission to publish this material under the Creative Commons Attribution 4.0 International (CC BY 4.0) License used by PLOS journals. Please closely review the details of PLOSu2019s copyright requirements here: PLOS Licenses and Copyright. If you need to request permissions from a copyright holder, you may use PLOS's Copyright Content Permission form.

Potential Copyright Issues:

- Figure 1A.. Please confirm whether you drew the images / clip-art within the figure panels by hand. If you did not draw the images, please provide (a) a link to the source of the images or icons and their license / terms of use; or (b) written permission from the copyright holder to publish the images or icons under our CC BY 4.0 license. Alternatively, you may replace the images with open source alternatives. See these open source resources you may use to replace images / clip-art:

4) Please amend your detailed Financial Disclosure statement. This is published with the article. It must therefore be completed in full sentences and contain the exact wording you wish to be published.

**Reviewers' comments:**

Reviewer's Responses to Questions

**Comments to the Authors:**

Reviewer #1: The manuscript uses a dynamical-systems approach (kernel-DMD) to integrate multi-omics time-series data in two Clusia species, revealing causal network drivers of C3 versus CAM photosynthesis and identifying biomarkers and control strategies relevant for crop bioengineering.

The authors have done substantial work in the area of multiome network integration. However, my primary concerns are as follows:

1. The authors should briefly introduce the specific strategy used, i.e., the components of the proposed kernel-DMD framework, rather than only emphasizing its significance. The significance may be included, but it should relate to the contribution and importance of kernel-DMD itself.

2. The Introduction and Methods sections are currently merged, which results in high cognitive load when reading. I strongly recommend separating them into two independent sections. Additionally, when presenting the Methods, a top-down structure should be used, starting with an overview followed by detailed subsections.

3. Why is the sigmoid kernel more suitable than Gaussian or polynomial kernels for multi-omics dynamic modeling? Is there any risk of overfitting? I recommend adding experimental comparison rather than relying solely on theoretical derivations to support this choice.

4. Equation (13) involves multiple objectives, but the manuscript does not sufficiently explain how the weights are determined or how they influence the optimization outcomes.

5. Using HSV to rank controllability and observability is meaningful, but questions remain regarding the stability of HSV. Does the feature list converge across iterations? A more thorough explanation or sensitivity analysis would strengthen this part.

6. The keywords need to be consistent throughout the entire manuscript. For example, but not limited to, “multi-ome network integration” and “multiome network integration.” Please check the wording throughout the manuscript and revise accordingly.

7. Although the authors provide a Zenodo repository for data and code availability, the files appear to be marked as Restricted. A publicly accessible GitHub repository is recommended for reproducibility.

8. Given the manuscript’s relevance to network science, the authors are encouraged to cite the following paper(s) to further strengthen the background and context.

DOI:10.1038/s41551-024-01312-5, DOI:10.1109/TSMC.2025.3578348, DOI:10.1109/TSMC.2025.3572738

Reviewer #2: This is a high quality, interesting and relatively novel study. I commend the authors on their work.

Reviewer #3: Dear authors and editor,

I have carefully read the manuscript entitled “Kernel-DMD with control for phenocopying of complex biological systems,” in which you describe and apply kernel dynamical mode decomposition to uncover photosynthetic dynamics from multiome data in Clusia major and Clusia rosea. I find this work potentially valuable to the biology community. However, in its current form, and from the perspective of someone with a modeling and statistical analysis background (though not an expert in DMD methods), I found the manuscript rather difficult to follow. Additionally, the overall purpose of the paper is not clearly articulated. While you demonstrate that the analysis yields insights into the photosynthetic processes of Clusia, and I understand that you also intend to showcase the broader applicability of the method, the presentation is not sufficiently explanatory. As a result, the manuscript is not easily accessible to readers outside the immediate field. I also believe that the manuscript requires substantial polishing in its organization, flow, and figures.

Regarding the introduction, parts of it read smoothly, but others feel out of place (see specific comments below). More importantly, the introduction does not adequately prepare the reader for the analyses that follow, nor does it explain why certain methodological choices are made later in the paper. In the description of the approaches used in the results, it would be very helpful to provide clearer motivation and intuition for the analytical decisions you make and to explain why these steps are necessary. Furthermore, key methodological components are intermixed with more technical or secondary details. I would recommend moving some of the more involved technicalities to supplementary material or to designated subsections in the Methods, allowing the main text to focus on conceptual clarity.

I also found the merged Results and Discussion section difficult to navigate. It becomes unclear where the presentation of results ends and where broader interpretation begins, particularly since the conclusion appears to begin around line 751 within a section titled “Sensitivity gain matrix identifies glycolytic enzymes as points of control.” It is fine to discuss results immediately after presenting them, but I would still suggest including a dedicated general discussion section to synthesize the main findings and their broader implications.

Several figures require improvement. Many are missing axis labels (e.g., Figures 3, 4, and 5 lack x-labels; Figure 8 lacks a color legend), and Figures 4 and 7 appear to be missing units for the decomposed dynamics of malic acid, H₂O, etc. Clearer figure presentation is essential for readability.

Specific comments:

Abstract: “We enhance the applicability of the Koopman operator to multiome integration.”

The meaning of enhance is unclear. Perhaps enable or demonstrate would be more precise.

Line 58: You may consider supplementing “vector that characterizes the state of the system” with intuition—for example, that could represent the abundance of molecular components.

Line 81: “Data-driven approaches ((17), (18), (19)) have emerged…”

I find the rest of this paragraph distracting from the main point and the flow. I would suggest simplifying it and focusing directly on the conceptual gap your work addresses.

Line 328: “Within the sigmoidal kernel we also implemented the graph Laplacian for integration of a priori knowledge of molecular pathways…”

It is not entirely clear how this implementation works. My assumption is that the graph Laplacian imposes constraints on allowable connectivity, but this should be clarified.

Line 350: The dash/apostrophe in the text differs from the symbol used in the corresponding equation.

Line 345: “We map inputs to physiological outputs Y in the following multi-objective function…”

I understand the intention here, but it would be useful to elaborate somewhere on the reasoning behind this formulation. As far as I understand, you aim to map X to Y and then interpret the DMD results to identify what drives the phenotypic outputs. Clarifying that these outputs quantify phenocopies would also help.

Line 364: “Linear operator from Taylor series expansion.”

This feels out of place. Please motivate why this expansion is introduced and how it relates to the main methodological narrative.

Closing remarks:

In summary, I believe the underlying work has value, and the methodology has the potential to be useful to the broader community. However, in its current form, the manuscript requires major rewriting to make it accessible, coherent, and reader-friendly. With substantial improvements to organization, clarity of exposition, motivation of methodological choices, and figure presentation, the paper could be significantly strengthened.

Reviewer #4: The study uses kernel-DMD to modeling photosynthesis. However, several issues regarding clarity, organization, notations, and biological interpretation need to be addressed before the manuscript can be considered for publication. My detailed comments are as follows:

1. The title uses the word "complex biological systems" is so big.

2. Update the references or revise the format, such as [16], [20].

3. Please check the word "multiome" and "multi-omics"

4. Figure 1 requires revision for clarity and readability.

5. Figure 2 illustrates photosynthesis pathways, which are not central to the methodological framework or the main results. I recommend moving this figure to the Supplementary Information.

6. The main text is heavily interwoven with biological explanations and mathematical derivations. This mixture obscures the key contributions and makes it difficult for readers to follow the main arguments. I suggest restructuring the text so that the mathematical framework and biological interpretations are presented more clearly and separately.

7. Line 146: In Eq. (2), please check and correct the notation for x_{t+1}

8. Line 170: The term “POD” is used without definition. Please provide a clear explanation upon first appearance.

9. Line 259: In Eq. (8), should the symbol on the right-hand side be "X" as in Eq.(4)?

10. Line 266: Why is "alpha =3" chosen? Please discuss the rationale and whether alternative parameter choices affect the results.

11. Line 341: The subsection number appears incorrect (“3.1.3”?). Please revise.

12.Line 375: Is X_j treated as fixed or variable in this context? Please clarify.

13. Line 381: The manuscript refers to "section 3.2," but this section is missing or mislabeled.

14. Line 398: "with 0 standard deviations" is that true? or just near 0?

15. The biological meaning of the different modes in Figures 3 and 4 is unclear. Please provide more explanation or biological context.

16. Figure 5 presents eigenmode topology, but the interpretation is difficult for readers. Additional explanation or annotation is needed.

17. The integration of the three omics layers (transcriptome, proteome, metabolome) appears somewhat artificial. Please elaborate on how their functional relationships are captured or modeled.

18. The concept of “control” in the system is not clearly defined. What does "control" biologically represent? Are there experimental validations for the model predictions, or are the results purely theoretical? How should readers interpret the reported "predictive accuracy"?

**Have the authors made all data and (if applicable) computational code underlying the findings in their manuscript fully available?**

The PLOS Data policy requires authors to make all data and code underlying the findings described in their manuscript fully available without restriction, with rare exception (please refer to the Data Availability Statement in the manuscript PDF file). The data and code should be provided as part of the manuscript or its supporting information, or deposited to a public repository. For example, in addition to summary statistics, the data points behind means, medians and variance measures should be available. If there are restrictions on publicly sharing data or code —e.g. participant privacy or use of data from a third party—those must be specified.requires authors to make all data and code underlying the findings described in their manuscript fully available without restriction, with rare exception (please refer to the Data Availability Statement in the manuscript PDF file). The data and code should be provided as part of the manuscript or its supporting information, or deposited to a public repository. For example, in addition to summary statistics, the data points behind means, medians and variance measures should be available. If there are restrictions on publicly sharing data or code —e.g. participant privacy or use of data from a third party—those must be specified.

Reviewer #1: None

Reviewer #2: Yes

Reviewer #3: Yes

Reviewer #4: None

PLOS authors have the option to publish the peer review history of their article (what does this mean? ). If published, this will include your full peer review and any attached files.). If published, this will include your full peer review and any attached files.

**Do you want your identity to be public for this peer review?** For information about this choice, including consent withdrawal, please see our For information about this choice, including consent withdrawal, please see our Privacy Policy ..

Reviewer #1: No

Reviewer #2: No

Reviewer #3: No

Reviewer #4: No

**Figure resubmission:**

**Reproducibility:**



---

## [Decision Letter · Decision Letter 1]

16 Feb 2026

Dear Ms Pierides,

We are pleased to inform you that your manuscript 'Kernel-DMD for multiome data integration and control' has been provisionally accepted for publication in PLOS Computational Biology.

Best regards,

Julio R. Banga, Ph.D.

Academic Editor

PLOS Computational Biology

Pedro Mendes

Section Editor

PLOS Computational Biology

Reviewer's Responses to Questions

**Comments to the Authors:**

Reviewer #1: All of my concerns have been addressed.

Reviewer #4: The authors have replied my questions, and I have no more comments.

**Have the authors made all data and (if applicable) computational code underlying the findings in their manuscript fully available?**

The PLOS Data policy requires authors to make all data and code underlying the findings described in their manuscript fully available without restriction, with rare exception (please refer to the Data Availability Statement in the manuscript PDF file). The data and code should be provided as part of the manuscript or its supporting information, or deposited to a public repository. For example, in addition to summary statistics, the data points behind means, medians and variance measures should be available. If there are restrictions on publicly sharing data or code —e.g. participant privacy or use of data from a third party—those must be specified.requires authors to make all data and code underlying the findings described in their manuscript fully available without restriction, with rare exception (please refer to the Data Availability Statement in the manuscript PDF file). The data and code should be provided as part of the manuscript or its supporting information, or deposited to a public repository. For example, in addition to summary statistics, the data points behind means, medians and variance measures should be available. If there are restrictions on publicly sharing data or code —e.g. participant privacy or use of data from a third party—those must be specified.

Reviewer #1: None

Reviewer #4: None

PLOS authors have the option to publish the peer review history of their article (what does this mean? ). If published, this will include your full peer review and any attached files.). If published, this will include your full peer review and any attached files.

**Do you want your identity to be public for this peer review?** For information about this choice, including consent withdrawal, please see our For information about this choice, including consent withdrawal, please see our Privacy Policy ..

Reviewer #1: No

Reviewer #4: No

---

## [Editor Report · Acceptance letter]

PCOMPBIOL-D-25-02160R1

Kernel-DMD for multiome data integration and control

Dear Dr Pierides,

I am pleased to inform you that your manuscript has been formally accepted for publication in PLOS Computational Biology. Your manuscript is now with our production department and you will be notified of the publication date in due course.

With kind regards,

Anita Estes
